



# Symmetric Equations on the Surface of a Sphere as Used by Model GISS:IB

Gary L. Russell[1], David H. Rind[1], and Jeffrey Jonas[2]

[1]NASA Goddard Institute for Space Studies, 2880 Broadway, New York, NY 10025, USA
[2]Center for Climate Research, Columbia University

**Correspondence:** Gary.L.Russell@nasa.gov

**Abstract.** Standard vector calculus formulas of Cartesian three space are projected onto the surface of a sphere. This produces symmetric equations with three nonindependent horizontal velocity components. Each orthogonal axis has a velocity component that rotates around its axis (eastward velocity rotates around the north-south axis) and a specific angular momentum component that is the product of the velocity component times the cosine of axis' latitude. Angular momentum components

align with the fixed axes and simplify several formulas, whereas the rotating velocity components are not orthogonal and vary with location. Three symmetric coordinates allow vector resolution and calculus operations continuously over the whole spherical surface, which is not possible with only two coordinates. The symmetric equations are applied to one-layer shallow water models on cubed-sphere and icosahedral grids, the latter being computationally simple and applicable to an ocean domain. Model results are presented for three different initial conditions and five different resolutions.

**1 Introduction**

According to the "hairy ball theorem" of Poincaré (proved by Brouwer) every continuous horizontal vector field on the surface of a sphere must have a **0**; a unit vector field must have a discontinuity. A differentiable coordinate on the surface of a sphere, for example latitude or longitude, will have a gradient unit vector that is tangent to the sphere; such a coordinate will also have a discontinuity. A horizontal vector component has a magnitude and a direction based on an underlying coordinate (e.g. eastward

velocity and longitude). The coordinate will have a discontinuity somewhere, so if the component is to be continuous over the whole sphere, its magnitude must be 0 where the coordinate is discontinuous. Scalar quantities have no associated direction and may be continuous over the whole sphere. Spatial derivatives acting on scalar quantities may use local coordinates, and coordinates of a point need not be continuous with respect to those of an adjacent point. Spatial derivatives acting on vector components require the components to be continuous. Coordinate discontinuities occur at the poles on a latitude-longitude

(lat-lon) grid or at the face edges on a cubed-sphere grid if coordinates are switched.

Many numerical schemes for solving the fluid dynamic equations on the surface of a sphere use two independent horizontal velocity components aligned with underlying coordinates. If the components are orthogonal such as on a lat-lon grid, then polar singularities may occur as well as other problems discussed in the Introduction of Heikes and Randall (1995). If the components are not orthogonal, then greater obtuseness of the components' angle decreases stability and precision of the



results. In addition, formulas involving the two coordinates are often not symmetric which means the flow is not isodirectional; this is certainly the case for standard lat-lon schemes (Williamson et al., 1992) or spectral harmonic schemes (Temperton, 1991); (Swarztrauber, 1993).

To use the lat-lon grid, but to avoid its polar deficiencies, the HYCOM ocean model (Sun and Bleck, 2005) uses a Mercator
lat-lon grid south of 58°N, and a different polar projection north of it, introducing the necessity to match variables along the boundary. Weller et al. (2012) tested a "Skipped lat-lon" grid that uses fewer cells in longitude poleward of 66° in a one-layer model. Other researchers have explored cubed-sphere grids: Adcroft et al. (2004), Putman and Lin (2007). The advantages of using a cubed-sphere grid over a lat-lon grid are that the two polar singularities on the lat-lon grid are replaced by eight weaker corner singularities on the cubed-sphere grid and the extreme aspect ratio of grid cells near the poles on a lat-lon
grid is eliminated as well as the polar filter used to increase the time step. A more thorough discussion of advantages and disadvantages of a cubed-sphere grid over a lat-lon grid occurs in Putman and Lin. Recent research on one-layer models has also been directed at icosahedral grids: Heikes and Randall (1995), Stuhne and Peltier (1999), Läuter et al. (2008), Lee and MacDonald (2009), Ringler et al. (2010), Weller et al. (2012), and others.

If two-component velocity is the prognostic transport variable that is advected in flux form, then spatial derivatives of vector
components will cause discontinuities to occur. This is a principal reason why researchers developed forms of the shallow water equations wherein scalar quantities such as potential vorticity, specific kinetic energy, and divergence are continuous everywhere. Computations are performed on local spherical coordinates or on the local tangent plane after which the horizontal velocity components are resurrected or time integrated by manipulating spatial derivatives of scalar quantities; spatial derivatives of vector components are not needed. Such forms include vector-invariant (Ringler et al., 2010), vorticity-divergence
(Williamson et al., 1992), or stream function and velocity potential (Masuda and Ohnishi, 1986), but the equations and programming can be complex.

The approach here uses three coordinates on the surface of a sphere to represent two-dimensional flow. Horizontal velocity is resolved by the three components, one for each of the mutually orthogonal axes. One component is eastward velocity and its coordinate is longitude which rotates around the north-south axis. The other two components rotate around the two equatorial
axes. Symmetric versions of the shallow water equations and vector calculus formulas are presented in which each coordinate has symmetric representation in the equations and formulas. Each coordinate is defined continuously except at its two poles. But, as a coordinate approaches one of its poles, its numerical importance in the formulas decreases, having no importance at the pole, and the other two coordinates become more nearly perpendicular. Consequently, vector resolution and calculus operations are continuous over the whole sphere including the poles. The symmetric equations are used to develop one-layer
shallow water equations models, one for a gnomonic cubed-sphere grid (CSK), one for an icosahedral B-grid with momentum defined at the primary grid cell corners (IB), and one for an icosahedral grid with a Voronoi tessellation (IK).

Several formulas of the symmetric equations are simplified by using relative specific angular momentum on the unit sphere, $\mathbf{A}$, as opposed to using the three velocity components. $\mathbf{A}$ is continuous everywhere; each component converges to 0 at its respective poles. $\mathbf{A}$ has been used by Ringler et al. (2010) and other researchers under the name $\mathbf{u}^{\perp}$, but it was not recognized as
the specific angular momentum vector on the surface of a sphere. The north-south axis component of $\mathbf{A}$ is identical to eastward



$u\cos\phi$ in the spectral model of Swarztrauber (1993) which also uses northward $v\cos\phi$. Both components are 0 and are continuous at the poles, but the other components of angular momentum are absent. Swarztrauber (1996) presents a spectral transform three-dimensional Cartesian method to solve the shallow-water equations on the sphere written in vorticity-divergence form. Putting aside the spectral transform method and vorticity-divergence form, there are some similarities and major differences.

Swarztrauber's equations exist in $\mathbf{R}^3$ and are later restricted to the spherical surface whereas symmetric equations are compressed to the surface from the beginning, and his equations use velocity instead of specific angular momentum.

The shallow water equations based on $\mathbf{A}$ are simpler than those using velocity or those using vector-invariant methods. Components of $\mathbf{A}$ are symmetric; polar problems are absent; and the metric term disappears from the momentum equation that exists when using eastward and northward velocity. Conservation of $\mathbf{A}$ by advection in flux form is precise without time

truncation errors. The symmetric equations are performed on the spherical surface without relying on tangent plane computations. Horizontal velocity and related $\mathbf{A}$ must be tangent to the spherical surface that allows only two degrees of freedom. Thus, three momentum components are not independent; there is a required alignment. If some process disturbs this alignment (e.g. advection or non-horizontal acceleration) causing momentum to no longer be horizontal, then a simple algorithm brings the three components back into alignment. This problem was recognized by Coté (1988) who added a "Lagrange multiplier"

term to the velocity equation of Cartesian $\mathbf{R}^3$ that was restricted to the spherical surface.

Computers require grid representations of differential equations; this causes numerical errors that relate to grid imprinting, mass variations, time integration, etc. Grid imprinting is easily recognized when integrating Williamson's [1992] solid body rotation Test Case 2, which lacks bottom topography. Errors due to grid imprinting should and do decrease with finer resolution.

Mass variations cause advection errors in numerical models and cause grid-matched alternating patterns. Mass is usually

conserved by programming advection to use flux form, but when mass is needed at different locations, it is specific mass or concentration that is interpolated. Tracers that follow mass advection include linear momentum and velocity, angular momentum and specific angular momentum, kinetic energy and specific kinetic energy, or absolute vorticity and potential vorticity. Russell and Lerner (1981) investigated mass variations and stated, "Tracer concentration is defined relative to air mass and not relative to space. In fact, with nonuniform mass, second- and fourth-order schemes become first-order schemes ..." The mass

coordinate at a point, in a spatial one-dimensional model, is the number of kilograms between the origin and the point. Second-order, fourth-order, and spectral schemes, in one dimension, usually compute their polynomial or sine wave coefficients from equally-distributed points in space, but the points are not equally-distributed in the mass coordinate, and consequently the derived coefficients are erroneous.

If tracer concentration is a linear function of mass over several grid cells in one dimension, then "the linear upstream

scheme" of Leer (1977) will perform the advection over those cells perfectly, even with arbitrary mass variations. Similarly, if concentration is a quadratic function of mass in one dimension, then "the quadratic upstream scheme" of Prather (1986) performs advection perfectly. Each of these schemes use mean tracer values and prognostic tracer gradients (and second order moments) inside each grid cell; these schemes are less sensitive to mass variations than are non-upstream schemes. The one-layer icosahedral model IB, discussed in detail later in this paper, uses a combination of linear upstream and second-order





schemes for momentum advection, but does not carry prognostic gradients. According to Weller et al. (2012), "an upwind-based interpolation of the potential vorticity controls the computational Rossby modes" in some one-layer models.

To be applied to the Earth, models should be tested with mass variations comparable to those on Earth. In some mountainous regions, surface pressure gradients and mass variations increase with finer resolution and so do their errors. Test Case 5 of

Williamson et al. (1992) is insufficient to judge model quality considering the topography variations that occur on Earth. Surface pressure gradients in Williamson do not vary with resolution and are two orders of magnitude smaller than the gradient at some locations on Earth, particularly the Andes. When a one-layer model is tested with various resolutions, it should include tests where such increased gradients exist. If a one-layer model requires special filtering to produce good results with an Earth topography simulation, then that filtering should be used when the model is applied to the less rigorous tests of Williamson.

Section 2 explains symmetric mathematics, including coordinates and variables, symmetric calculus operators, and the differential solution to the shallow water equations in terms of symmetric coordinates. Section 3 presents the discrete implementation of alignment, the pressure gradient force, advection, the Coriolis force, and other aspects of icosahedral models including grid arrangements and time steps. Section 4 applies 3 test cases to lat-lon, cubed-sphere, and icosahedral one-layer models with various resolutions. Section 5 contains discussion and conclusions.

Unlike computer languages, division has lower precedence than does multiplication in this paper. Vector quantities are indicated by bold capital letters; when displayed by three coordinates, e.g. $\mathbf{A} = (a, b, c)$, they are the Cartesian coordinates of $\mathbf{R}^3$ and not a local coordinate system.

## 2 Symmetric mathematics

### 2.1 Coordinates and variables

A sphere of radius 1 is centered at the origin in three-dimensional Cartesian space with axis unit vectors $\mathbf{X} = (1, 0, 0)$, $\mathbf{Y} = (0, 1, 0)$, and $\mathbf{Z} = (0, 0, 1)$. A concentric sphere represents the fixed Earth with its radius and its geometric labels such as latitude, longitude, equator, and poles. The $\mathbf{Z}$ axis is aligned with the Earth's north-south axis, but this is only necessary for simplifying the sine of latitude for the Coriolis force. Cartesian coordinates on the surface of the unit sphere are labeled $\mathbf{P} = (p, q, r)$ where $p^2 + q^2 + r^2 = 1$; this label is also used for points on the Earth's surface, but they mean the projection onto the unit sphere.

Note that $r$ and $\sqrt{p^2 + q^2}$ are sine and cosine of Earth latitude.

Three horizontal velocity components, $u$, $v$ and $w$ (m/s), defined everywhere on the Earth's surface, except at their poles, rotate around each respective axis. Velocity unit vectors of the three symmetric components and the northward velocity com-





ponent $n$, at point $\mathbf{P}$, are

$$\mathbf{U} = \frac{\mathbf{X} \times \mathbf{P}}{|\mathbf{X} \times \mathbf{P}|} = \frac{(0, -r, q)}{\sqrt{q^2 + r^2}}; \tag{2.1}$$

$$\mathbf{V} = \frac{\mathbf{Y} \times \mathbf{P}}{|\mathbf{Y} \times \mathbf{P}|} = \frac{(r, 0, -p)}{\sqrt{r^2 + p^2}}; \tag{2.2}$$

$$\mathbf{W} = \frac{\mathbf{Z} \times \mathbf{P}}{|\mathbf{Z} \times \mathbf{P}|} = \frac{(-q, p, 0)}{\sqrt{p^2 + q^2}}, \quad \text{which points eastward;} \tag{2.3}$$

$$\mathbf{N} = \mathbf{P} \times \mathbf{W} = \frac{(-rp, -qr, p^2 + q^2)}{\sqrt{p^2 + q^2}}, \quad \text{which points northward.} \tag{2.4}$$

$\mathbf{S}$ (m/s) is horizontal velocity at point $\mathbf{P}$ and is equal to $\mathbf{u}$ of Cartesian $\mathbf{R}^3$ in Ringler et al. [2010]. Since $u = \mathbf{S} \cdot \mathbf{U}$, $v = \mathbf{S} \cdot \mathbf{V}$, and $w = \mathbf{S} \cdot \mathbf{W}$, then

$$\mathbf{S} = (rv\sqrt{r^2 + p^2} - qw\sqrt{p^2 + q^2}, pw\sqrt{p^2 + q^2} - ru\sqrt{q^2 + r^2}, qu\sqrt{q^2 + r^2} - pv\sqrt{r^2 + p^2}). \tag{2.5}$$

Although there are three components, there are only two degrees of freedom since $\mathbf{S} \cdot \mathbf{P} = 0$.

Relative specific angular momentum on the unit sphere, $\mathbf{A}$ (m/s), uses the variables $(a, b, c) = (u\sqrt{q^2 + r^2}, v\sqrt{r^2 + p^2}, w\sqrt{p^2 + q^2})$. The component $c$ pointing toward the $\mathbf{Z}$ axis equals eastward velocity times the cosine of Earth latitude. If $\mathbf{A}$ and $\mathbf{S}$ are properly aligned, i.e. horizontal or tangent to the spherical surface at $\mathbf{P}$, then they have the relationships $\mathbf{A} \cdot \mathbf{P} = \mathbf{S} \cdot \mathbf{P} = 0$, $\mathbf{A} = \mathbf{P} \times \mathbf{S}$, and $\mathbf{S} = \mathbf{A} \times \mathbf{P}$. Thus, on the surface of the sphere, $\mathbf{A}$ is at right angle to $\mathbf{S}$ and both are tangent to the surface. The components of $\mathbf{A}$ are mutually orthogonal being aligned with the $\mathbf{X}$, $\mathbf{Y}$ and $\mathbf{Z}$ axes that are fixed with respect to the Earth, whereas the components of horizontal velocity aligned with the unit vectors $\mathbf{U}$, $\mathbf{V}$ and $\mathbf{W}$ are not orthogonal and change with location. Also, $a$, $b$ and $c$ are continuous everywhere, whereas $u$, $v$ and $w$ are discontinuous at their respective poles. Formulas that simplify $\mathbf{A}$ over velocity are

$$\mathbf{S} = (rb - qc, pc - ra, qa - pb), \tag{2.6}$$

or the northern velocity component $n = \mathbf{S} \cdot \mathbf{N} = (qa - pb)/\sqrt{p^2 + q^2}$. The velocity component rotated $90°$ from $\mathbf{U}$ is $\mathbf{S} \cdot \mathbf{P} \times \mathbf{U} = (rb - qc)/\sqrt{q^2 + r^2}$ and that rotated from $\mathbf{V}$ is $\mathbf{S} \cdot \mathbf{P} \times \mathbf{V} = (pc - ra)/\sqrt{r^2 + p^2}$ which are used for the Coriolis force. Velocity squared is $\mathbf{S} \cdot \mathbf{S} = \mathbf{A} \cdot \mathbf{A} = a^2 + b^2 + c^2$.

Spherical angular rotation coordinates, measured in radians, that rotate around the $\mathbf{X}$, $\mathbf{Y}$ and $\mathbf{Z}$ axes are $\mu$, $\nu$ and $\lambda$, respectively. Angular rotation coordinates measured from pole to pole for each axis are $\delta$, $\epsilon$ and $\phi$, respectively. $\lambda$ and $\phi$ are Earth longitude and latitude. A point on the sphere can be designated by any of four different coordinate systems:

$$(p, q, r) = (\sin \delta, \cos \mu \cos \delta, \sin \mu \cos \delta) =$$

$$= (\sin \nu \cos \epsilon, \sin \epsilon, \cos \nu \cos \epsilon) =$$

$$= (\cos \lambda \cos \phi, \sin \lambda \cos \phi, \sin \phi). \tag{2.7}$$

At any point, the gradients of $\mu$, $\nu$, $\lambda$ and $\phi$ on the spherical surface are parallel to the unit vectors $\mathbf{U}$, $\mathbf{V}$, $\mathbf{W}$ and $\mathbf{N}$. Partial derivatives of the angular rotation coordinates with respect to one another are needed to derive new and old forms of various





terms. A change in $\Delta\phi$ causes a change in $\Delta\mu\cos\delta$ in the ratio of $\mathbf{U}\cdot\mathbf{N}$. In the limit, $\partial\mu/\partial\phi = \mathbf{U}\cdot\mathbf{N}/\cos\delta$. A few useful derivatives are:

$$\frac{\partial\lambda}{\partial\mu} = \frac{\mathbf{W}\cdot\mathbf{U}\cos\delta}{\cos\phi} = \frac{-rp}{p^2+q^2} = -\cos\lambda\tan\phi; \tag{2.8}$$

$$\frac{\partial\phi}{\partial\mu} = \mathbf{N}\cdot\mathbf{U}\cos\delta = \frac{q}{\sqrt{p^2+q^2}} = \sin\lambda; \tag{2.9}$$

$$\frac{\partial\mu}{\partial\lambda} = \frac{\mathbf{U}\cdot\mathbf{W}\cos\phi}{\cos\delta} = \frac{-rp}{q^2+r^2}; \tag{2.10}$$

$$\frac{\partial\mu}{\partial\phi} = \frac{\mathbf{U}\cdot\mathbf{N}}{\cos\delta} = \frac{q}{(q^2+r^2)\sqrt{p^2+q^2}} \tag{2.11}$$

Variables for the shallow water equations on the sphere are the height field above the surface topography, $h$ (m), and $\mathbf{A}$. Because density (kg/m$^3$) is uniform and is set to 1, $h$ and mass per unit area are used interchangeably. The surface topography, $h_S$ (m), is specified. $R$ (m), the Earth's radius, $g$ (m/s$^2$), the downward vertical acceleration due to gravity, and $\Omega$ (1/s), the Earth's angular rotation rate, are assumed to be uniform. The field top geopotential $\Phi$ (m$^2$/s$^2$) $= g(h + h_S)$.

The new symmetric equations to be presented here are applicable to many grid arrangements; one is the gnomonic cubed-sphere grid. A symmetric tessellation of the surface of a cube [-1:1, -1:1, -1:1] by line segments of length 2 parallel to an edge projects a rectangular grid onto great circle arcs on the surface of the sphere. Faces of the cube are numbered 1 to 6: $z = 1$, $y = -1$, $x = 1$, $z = -1$, $y = 1$, $x = -1$. See Fig. 1. On the sphere, grid cells are shaped like parallelograms and at least one of the unit vectors, $\mathbf{U}$, $\mathbf{V}$ or $\mathbf{W}$, will be perpendicular to any grid edge. It was this property that lead to the exploration of symmetric equations. At corners of the cube, velocity unit vectors are separated by 60 or 120°. A proprietary one-layer shallow water equations model, labeled CSK, may be published later, but some results are presented in Section 4.

Two new shallow water equations models were developed on icosahedral grids. The centers of primary grid cells are the vertices of a triangular lattice covering the sphere; the cells themselves are pentagons and irregular hexagons. The B-grid icosahedral model, labeled IB, is discussed in detail in this paper. Its momentum cells (described as the dual mesh by Ringler et al. (2010)) are spherical triangles whose vertices are three primary cell centers and whose momentum centers are primary cell corners. The second icosahedral model, IK, uses proprietary techniques similar to CSK. All three models use the symmetric equations, but IB and IK perform less quantity averaging than CSK does with its unusual grid cell shape. Figure 2 shows a four triangle wedge of an icosahedral grid.

## 2.2 The symmetric $\nabla_S$ operator on the surface of a sphere

The symmetric del operator on the surface of a sphere $\nabla_S$, or simply $\nabla$ in this paper, is

$$\nabla = \left(\cos\delta\frac{\partial}{\partial\delta}, \cos\epsilon\frac{\partial}{\partial\epsilon}, \cos\phi\frac{\partial}{\partial\phi}\right)/R =$$
$$= \left(r\frac{\partial}{\partial\nu} - q\frac{\partial}{\partial\lambda}, p\frac{\partial}{\partial\lambda} - r\frac{\partial}{\partial\mu}, q\frac{\partial}{\partial\mu} - p\frac{\partial}{\partial\nu}\right)/R \tag{2.12}$$

which is equivalent to the common two-dimensional del operator on the surface of a sphere, $\nabla_R = (\mathbf{W}\partial/\partial\lambda + \mathbf{N}\cos\phi\partial/\partial\phi)/R\cos\phi$ [Williamson et al., 1992, Eq. 3]. $\nabla_S$ or $\nabla_R$ can be applied to three space, and be equivalent to the three-dimensional Cartesian





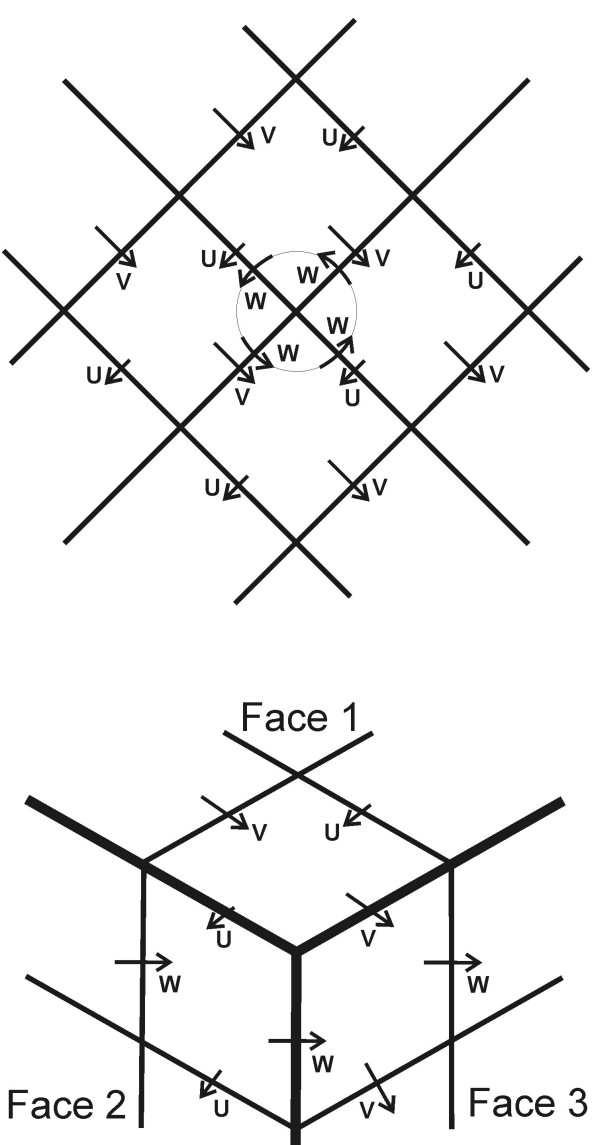

**Figure 1.** Upper diagram shows arrangement of velocity components on Face 1 of cubed-sphere grid centered around the North Pole. Lower diagram shows arrangement of velocity components at the intersection of Faces 1, 2, and 3. Although **U**, **V**, and **W** are defined everywhere except at their poles, only the component perpendicular to an edge is displayed.





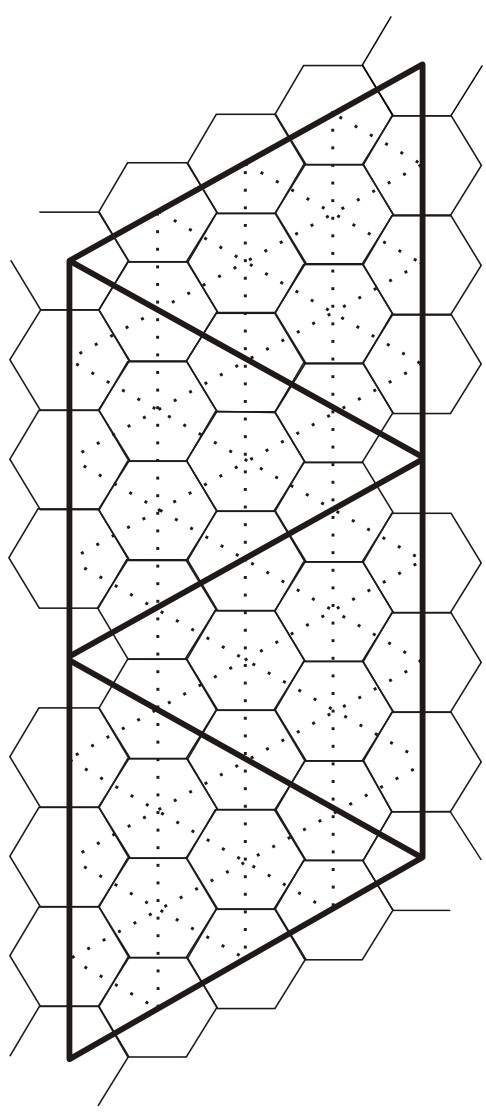

**Figure 2.** Four triangles of an icosahedron are surrounded by thick, bold lines. Vertices of the bold triangles are centers of incomplete pentagonal primary cells. Edges of primary cells, for grid level 2, are indicated by thin solid lines. Dotted lines, between primary cell centers, indicate edges of triangular momentum cells. Primary cell corners are momentum cell centers.



operator $\nabla_C = (\partial/\partial x, \partial/\partial y, \partial/\partial z)$, by restricting quantities to which they are applied to be radially constant, or by adding the term $\mathbf{P}\partial/\partial\rho$ to the del operator, where $\rho$ (m) is the radial coordinate. Computations of $\partial Q/\partial\lambda$ (or $\partial Q/\partial\phi$), an arbitrary scalar $Q$, are identical in formulas that use $\nabla_S$ or $\nabla_R$, but the factors that multiply those partial derivatives in the formulas are different.

5     The gradient of a scalar $h$ on the Earth's surface is

$$\nabla h = \left( \cos\delta\frac{\partial h}{\partial\delta}, \cos\epsilon\frac{\partial h}{\partial\epsilon}, \cos\phi\frac{\partial h}{\partial\phi} \right)/R =$$
$$= \left( r\frac{\partial h}{\partial\nu} - q\frac{\partial h}{\partial\lambda}, p\frac{\partial h}{\partial\lambda} - r\frac{\partial h}{\partial\mu}, q\frac{\partial h}{\partial\mu} - p\frac{\partial h}{\partial\nu} \right)/R =$$
$$= \left( \mathbf{U}\cos\delta\frac{\partial h}{\partial\mu} + \mathbf{V}\cos\epsilon\frac{\partial h}{\partial\nu} + \mathbf{W}\cos\phi\frac{\partial h}{\partial\lambda} \right)/R. \tag{2.13}$$

The gradient vector is tangent to the sphere at point $\mathbf{P}$, and, as $\mathbf{P}$ approaches the north or south pole, $r$ approaches $\pm 1$; $\cos\delta$ and $\cos\epsilon$ approach 1; $p$, $q$, and $\cos\phi$ approach 0; $\partial h/\partial\phi$ and $\partial h/\partial\lambda$ become less important in Eq. 2.13; and $\mathbf{U}$ and $\mathbf{V}$ become more nearly perpendicular. Using perpendicular unit vectors $\mathbf{W}$ and $\mathbf{N}$, the gradient is commonly written as

10     $\nabla h = (\mathbf{W}\partial h/\partial\lambda + \mathbf{N}\cos\phi\partial h/\partial\phi)/R\cos\phi.$            (2.14)

This common form is not valid near the poles because of $\cos\phi$ in the denominator while the new symmetric form, Eq. 2.13, is valid everywhere, a benefit of using the symmetric equations. The equivalence between the common form and the new form of $\nabla h$ is shown in Appendix A. If $\nabla h$ is treated as a velocity, then the specific angular momentum with which it is associated is

$$\mathbf{A} = \mathbf{P} \times \nabla h = \left( \frac{\partial h}{\partial\mu}, \frac{\partial h}{\partial\nu}, \frac{\partial h}{\partial\lambda} \right)/R. \tag{2.15}$$

15     The divergence of a horizontal vector $\mathbf{D} = (d, e, f)$ at point $\mathbf{P}$ is

$$\nabla \cdot \mathbf{D} = \left( \cos\delta\frac{\partial d}{\partial\delta} + \cos\epsilon\frac{\partial e}{\partial\epsilon} + \cos\phi\frac{\partial f}{\partial\phi} \right)/R =$$
$$= \left[ \frac{\partial(d\cos\delta)}{\partial\delta} + \frac{\partial(e\cos\epsilon)}{\partial\epsilon} + \frac{\partial(f\cos\phi)}{\partial\phi} \right]/R =$$
$$= \left( r\frac{\partial d}{\partial\nu} - q\frac{\partial d}{\partial\lambda} + p\frac{\partial e}{\partial\lambda} - r\frac{\partial e}{\partial\mu} + q\frac{\partial f}{\partial\mu} - p\frac{\partial f}{\partial\nu} \right)/R =$$
$$= \left[ \frac{\partial(qf-re)}{\partial\mu} + \frac{\partial(rd-pf)}{\partial\nu} + \frac{\partial(pe-qd)}{\partial\lambda} \right]/R, \tag{2.16}$$

which is equivalent to the common form on the surface of a sphere:

$[\partial(\mathbf{D}\cdot\mathbf{W})/\partial\lambda + \partial(\mathbf{D}\cdot\mathbf{N}\cos\phi)/\partial\phi]/R\cos\phi$. The first two forms of Eq. 2.16 are equivalent because $\mathbf{D}\cdot\mathbf{P} = 0$ and

$$\frac{\partial(d\cos\delta)}{\partial\delta} + \frac{\partial(e\cos\epsilon)}{\partial\epsilon} + \frac{\partial(f\cos\phi)}{\partial\phi} =$$
$$= \cos\delta\frac{\partial d}{\partial\delta} - d\sin\delta + \cos\epsilon\frac{\partial e}{\partial\epsilon} - e\sin\epsilon + \cos\phi\frac{\partial f}{\partial\phi} - f\sin\phi =$$
$$= \cos\delta\frac{\partial d}{\partial\delta} + \cos\epsilon\frac{\partial e}{\partial\epsilon} + \cos\phi\frac{\partial f}{\partial\phi} - \mathbf{D}\cdot\mathbf{P}. \tag{2.17}$$





Similar reasoning shows the equivalence between the third and fourth forms of Eq. 2.16. If $\mathbf{D}$ is horizontal velocity, $\mathbf{D} = \mathbf{S}$, then the divergence of $\mathbf{S}$ can also be written as

$$\nabla \cdot \mathbf{S} = \nabla \cdot \mathbf{A} \times \mathbf{P} = \left( \frac{\partial a}{\partial \mu} + \frac{\partial b}{\partial \nu} + \frac{\partial c}{\partial \lambda} \right) / R. \tag{2.18}$$

Also,

$\quad \nabla \cdot \mathbf{A} = \nabla \cdot \mathbf{P} \times \mathbf{S} = -\nabla \cdot \mathbf{S} \times \mathbf{P} = -\nabla \cdot \mathbf{D} \times \mathbf{P} = -\left( \dfrac{\partial d}{\partial \mu} + \dfrac{\partial e}{\partial \nu} + \dfrac{\partial f}{\partial \lambda} \right) / R.$ $\qquad(2.19)$

Noting that $\nabla h$ is perpendicular to $\mathbf{P}$ and reasoning similar to Eq. 2.17, the Laplacian is

$$\nabla^2 h = \nabla \cdot (\nabla h) = \left( \cos^2 \delta \frac{\partial^2 h}{\partial \delta^2} + \cos^2 \epsilon \frac{\partial^2 h}{\partial \epsilon^2} + \cos^2 \phi \frac{\partial^2 h}{\partial \phi^2} \right) / R^2 =$$
$$= \left( \frac{\partial^2 h}{\partial \mu^2} + \frac{\partial^2 h}{\partial \nu^2} + \frac{\partial^2 h}{\partial \lambda^2} \right) / R^2. \tag{2.20}$$

The Laplacian of Eq. 2.20 is equivalent to the common form on the surface of a sphere:
$[\partial^2 h / \partial \lambda^2 + \cos \phi \, \partial (\cos \phi \, \partial h / \partial \phi) / \partial \phi] / R^2 \cos^2 \phi.$

$\quad$ The curl of a horizontal vector $\mathbf{D} = (d, e, f)$ is

$$\nabla \times \mathbf{D} = \left( \cos \epsilon \frac{\partial f}{\partial \epsilon} - \cos \phi \frac{\partial e}{\partial \phi}, \cos \phi \frac{\partial d}{\partial \phi} - \cos \delta \frac{\partial f}{\partial \delta}, \cos \delta \frac{\partial e}{\partial \delta} - \cos \epsilon \frac{\partial d}{\partial \epsilon} \right) / R. \tag{2.21}$$

If $\nabla \times \mathbf{D}$ were completely vertical, then $(\nabla \times \mathbf{D}) \times \mathbf{P}$ would be $\mathbf{0}$. It is not; in fact $(\nabla \times \mathbf{D}) \times \mathbf{P} = \mathbf{D}/R$. The upward vertical component of the curl of $\mathbf{D}$ is

$$\mathbf{P} \cdot \nabla \times \mathbf{D} = \left[ p \left( \cos \epsilon \frac{\partial f}{\partial \epsilon} - \cos \phi \frac{\partial e}{\partial \phi} \right) + q \left( \cos \phi \frac{\partial d}{\partial \phi} - \cos \delta \frac{\partial f}{\partial \delta} \right) + r \left( \cos \delta \frac{\partial e}{\partial \delta} - \cos \epsilon \frac{\partial d}{\partial \epsilon} \right) \right] / R$$
$$= \left( \frac{\partial d}{\partial \mu} + \frac{\partial e}{\partial \nu} + \frac{\partial f}{\partial \lambda} \right) / R = -\nabla \cdot \mathbf{P} \times \mathbf{D}, \tag{2.22}$$

$\quad$ which is equivalent to the common form on the surface of a sphere:
$[\partial (\mathbf{D} \cdot \mathbf{N}) / \partial \lambda - \partial (\mathbf{D} \cdot \mathbf{W} \cos \phi) / \partial \phi] / R \cos \phi.$ If $\mathbf{D} = \mathbf{S}$, then the vertical component of the curl of $\mathbf{S}$ can also be written as

$$\mathbf{P} \cdot \nabla \times \mathbf{S} = \mathbf{P} \cdot \nabla \times (\mathbf{A} \times \mathbf{P}) = -\nabla \cdot \mathbf{A} =$$
$$= -\left[ \frac{\partial (a \cos \delta)}{\partial \delta} + \frac{\partial (b \cos \epsilon)}{\partial \epsilon} + \frac{\partial (c \cos \phi)}{\partial \phi} \right] / R. \tag{2.23}$$

The upward vertical component of relative vorticity is

$$\zeta = \mathbf{P} \cdot \nabla \times \mathbf{S} = \mathbf{P} \cdot \nabla \times (\mathbf{A} \times \mathbf{P}) \tag{2.24}$$

$\quad$ which is equivalent to the common form on the surface of a sphere:
$[\partial n / \partial \lambda - \partial (w \cos \phi) / \partial \phi] / R \cos \phi.$

If $\mathbf{G}$ and $\mathbf{H}$ are differentiable vectors in three space, then a well known identity using the Cartesian del operator is

$$\mathbf{G} \cdot \nabla_C \times \mathbf{H} = \mathbf{H} \cdot \nabla_C \times \mathbf{G} - \nabla_C \cdot \mathbf{G} \times \mathbf{H}. \tag{2.25}$$





If $\mathbf{G}$ is radially aligned and of constant magnitude ($\mathbf{P}$ for example), then $\nabla_C \times \mathbf{G} = \mathbf{0}$. If $\mathbf{G}$ is a radially aligned unit vector and $\mathbf{H}$ is perpendicular to $\mathbf{G}$, then

$$\mathbf{G} \cdot \nabla_C \times (\mathbf{H} \times \mathbf{G}) = -\nabla_C \cdot \mathbf{G} \times (\mathbf{H} \times \mathbf{G}) = -\nabla_C \cdot \mathbf{H}. \tag{2.26}$$

The above relationships apply to the present $\nabla$ (or $\nabla_S$) operator and are shown in the relationships of Eqs. 2.22 and 2.23.

Many of the new symmetric forms have no varying quantities outside their derivatives and can be integrated using Green's Theorem. Proofs of several of the equivalences used above are available at https://aom.giss.nasa.gov .

### 2.3   Differential form of the shallow water equations

The differential form for conservation of mass, using Eq. 2.18, is applied to mass per unit area.

$$\frac{\partial h}{\partial t} = -\nabla \cdot (h\mathbf{S}) = -\nabla \cdot (h\mathbf{A} \times \mathbf{P}) = -\left[ \frac{\partial(ha)}{\partial \mu} + \frac{\partial(hb)}{\partial \nu} + \frac{\partial(hc)}{\partial \lambda} \right]/R =$$
$$= -\left[ \cos\delta \frac{\partial(hu)}{\partial \mu} + \cos\epsilon \frac{\partial(hv)}{\partial \nu} + \cos\phi \frac{\partial(hw)}{\partial \lambda} \right]/R. \tag{2.27}$$

This symmetric form is equivalent to the common equation:

$$\frac{\partial h}{\partial t} + \left[ \frac{\partial(hw)}{\partial \lambda} + \frac{\partial(hn\cos\phi)}{\partial \phi} \right]/R\cos\phi = 0. \tag{2.28}$$

The three-component advective form for specific angular momentum is

$$\frac{\partial \mathbf{A}}{\partial t} = -\left( a\frac{\partial \mathbf{A}}{\partial \mu} + b\frac{\partial \mathbf{A}}{\partial \nu} + c\frac{\partial \mathbf{A}}{\partial \lambda} \right)/R + f\mathbf{A} \times \mathbf{P} - \mathbf{P} \times \nabla\Phi \tag{2.29}$$

where $f$ (1/s) is the Coriolis parameter and $\Phi$ (m$^2$/s$^2$) is the fluid top geopotential:

$$f \;=\; 2\Omega\sin\phi; \tag{2.30}$$
$$\Phi \;=\; g(h + h_S). \tag{2.31}$$

Replacing $h$ with $h\mathbf{A}$ in Eqs. 2.27 and 2.28 shows the differential form for momentum advection in flux form:

$$\frac{\partial(h\mathbf{A})}{\partial t} + \left[ \frac{\partial(ha\mathbf{A})}{\partial \mu} + \frac{\partial(hb\mathbf{A})}{\partial \nu} + \frac{\partial(hc\mathbf{A})}{\partial \lambda} \right]/R =$$
$$= \frac{\partial(h\mathbf{A})}{\partial t} + \left[ \frac{\partial(hw\mathbf{A})}{\partial \lambda} + \frac{\partial(hn\mathbf{A}\cos\phi)}{\partial \phi} \right]/R\cos\phi =$$
$$= h\left[ \frac{\partial \mathbf{A}}{\partial t} + \left( w\frac{\partial \mathbf{A}}{\partial \lambda} + n\cos\phi\frac{\partial \mathbf{A}}{\partial \phi} \right)/R\cos\phi \right] +$$
$$+ \mathbf{A}\left\{ \frac{\partial h}{\partial t} + \left[ \frac{\partial(hw)}{\partial \lambda} + \frac{\partial(hn\cos\phi)}{\partial \phi} \right]/R\cos\phi \right\}. \tag{2.32}$$

The last line of Eq. 2.32 evaluates to zero because of conservation of mass. Inside the square brackets in the penultimate line

of Eq. 2.32 shows the form that may be used for advection of $\mathbf{A}$ at a grid edge when the edge is perpendicular to the unit vector





**W**. Replacing **A** in the square brackets in that penultimate line with $c = w \cos \phi$ and dividing by $\cos \phi$ yields:

$$\left\{ \frac{\partial (w \cos \phi)}{\partial t} + \left[ w \frac{\partial (w \cos \phi)}{\partial \lambda} + n \cos \phi \frac{\partial (w \cos \phi)}{\partial \phi} \right] / R \cos \phi \right\} / \cos \phi =$$
$$= \frac{\partial w}{\partial t} + \left( w \frac{\partial w}{\partial \lambda} + n \cos \phi \frac{\partial w}{\partial \phi} \right) / R \cos \phi - n w \tan \phi / R \qquad (2.33)$$

which are the common shallow water forms for the time derivative, the advective terms, and the metric term of eastward velocity. Application of the momentum conservation form to **A** thus includes the metric term. Although the common shallow

water form for $\partial w / \partial t$ can incorporate the metric term into the advective terms, $\partial n / \partial t$ cannot conveniently. In the present formulation, all three components act like $\partial c / \partial t$ with the metric term included into the advective terms.

Symmetric versions of the shallow water equations using vorticity and divergence or vector-invariant form are shown at https://aom.giss.nasa.gov . Model IB does not use this form nor Eq. 2.33.

## 3  Discrete implementation of symmetric equations

### 3.1  Alignment of velocity or specific angular momentum

Alignment of the three momentum components at **P** means that $\mathbf{A} = (u \cos \delta, v \cos \epsilon, w \cos \phi)$ and **S** are perpendicular to **P** ($\mathbf{A} \cdot \mathbf{P} = \mathbf{S} \cdot \mathbf{P} = 0$). Application of a horizontal acceleration vector to the components maintains alignment. Thus, the pressure gradient force vector and the Coriolis force maintain alignment. The pressure gradient force obtained via Green's Theorem and advection may distort alignment. The following procedure brings distorted momentum components back into alignment.

Given unaligned velocity components $u$, $v$ and $w$ at point **P**, determine the least square fit velocity vector $\mathbf{S}_{NEW}$ that is horizontal ($\mathbf{S}_{NEW} \cdot \mathbf{P} = 0$) and best matches the components weighted by the square of the distance to their axes.

$$s = \cos^2 \delta (\mathbf{S}_{NEW} \cdot \mathbf{U} - u)^2 + \cos^2 \epsilon (\mathbf{S}_{NEW} \cdot \mathbf{V} - v)^2 + \cos^2 \phi (\mathbf{S}_{NEW} \cdot \mathbf{W} - w)^2. \qquad (3.1)$$

If $u$, $v$ and $w$ were already aligned then $t = 0$, where

$$t = p u \cos \delta + q v \cos \epsilon + r w \cos \phi. \qquad (3.2)$$

When $s$ is minimized, alignment of distorted components $u$, $v$ and $w$ produces

$$u_{NEW} = \mathbf{S}_{NEW} \cdot \mathbf{U} = u - p t / \cos \delta; \qquad (3.3)$$
$$v_{NEW} = \mathbf{S}_{NEW} \cdot \mathbf{V} = v - q t / \cos \epsilon; \qquad (3.4)$$
$$w_{NEW} = \mathbf{S}_{NEW} \cdot \mathbf{W} = w - r t / \cos \phi. \qquad (3.5)$$

Performing this analysis with specific angular momentum components yields: $a = u \cos \delta$, $t = p a + q b + r c = \mathbf{P} \cdot \mathbf{A}$, and

$$\mathbf{A}_{NEW} = \mathbf{A} - \mathbf{P} (\mathbf{P} \cdot \mathbf{A}). \qquad (3.6)$$





The minimization technique applied above is equivalent to projecting an unaligned $\mathbf{S}$ or $\mathbf{A}$ onto the tangent plane of the spherical surface at $\mathbf{P}$. When $\mathbf{P}$ is close to an axis pole, say $r$ is close to $\pm 1$, then $\cos\phi$ is close to 0, $w$ or $c$ is most strongly modified, and $u$ and $v$ are modified weakly. Alignment has the same purpose as the "Lagrange multiplier" term of Coté (1988).

The benefit of using three aligned components instead of two for advection on a particular cubed-sphere grid is shown at
https://aom.giss.nasa.gov .

### 3.2   Pressure gradient force for model IB

Change of velocity by the pressure gradient force is proportional to the gradient of the field top geopotential $\Phi$:

$$\Delta\mathbf{S} = -\Delta t\nabla\Phi = -g\Delta t\nabla(h + h_S) \tag{3.7}$$

where $\Delta t$ is the time step. Change of specific angular momentum, using Eq. 2.15, is

$$\Delta\mathbf{A} = -\Delta t\mathbf{P}\times\nabla\Phi = -\Delta t\left(\frac{\partial\Phi}{\partial\mu}, \frac{\partial\Phi}{\partial\nu}, \frac{\partial\Phi}{\partial\lambda}\right)/R. \tag{3.8}$$

Application of the pressure gradient force to velocity averaged over an arc usually involves interpolating $\Phi$ to the corners of the arc and to $\Phi$ on either side of the arc. Application to velocity averaged over a cell is conveniently performed using Green's Theorem. This computation is discussed first for cubed-sphere models and later for icosahedral models, like IB.

Acceleration of velocity averaged over a primary cell of a cubed-sphere model starts by knowing $\Phi$ averaged over the cell's
edges from interpolation of primary cell values. For each cell mean velocity component, edge $\Phi$ multiplied by the cosine of the angle between the component's unit vector and a unit vector outwardly perpendicular to the edge is integrated with respect to distance around the cell's edges. This integral, multiplied by the time step and divided by the cell's area, accelerates the mean velocity component of the cell. On Face 1 (Fig. 1), where the outward perpendicular direction of the right and left edges are $\mathbf{V}$ and $-\mathbf{V}$ and that of the top and bottom edges are $-\mathbf{U}$ and $\mathbf{U}$ and where $d\chi_W = R\cos\phi d\lambda$ is the spatial differential in the
direction of $\mathbf{W}$:

$$\begin{aligned}
\Delta c &= \Delta w\cos\phi = -\Delta t\cos\phi\frac{\partial\Phi}{\partial\chi_W} = -\Delta t\frac{\partial(\Phi\cos\phi)}{\partial\chi_W} = \\
&= -\Delta t[(\Phi L\mathbf{W}\cdot\mathbf{V}\cos\phi)_{RIGHT} - (\Phi L\mathbf{W}\cdot\mathbf{V}\cos\phi)_{LEFT} - \\
&\quad - (\Phi L\mathbf{W}\cdot\mathbf{U}\cos\phi)_{TOP} + (\Phi L\mathbf{W}\cdot\mathbf{U}\cos\phi)_{BOTTOM}]/K
\end{aligned} \tag{3.9}$$

where $L$ is the arc length of an edge and $K$ is the primary cell area [I-1:I, J-1:J] in the computer implementation. For the right edge [I, J-1:J], $L\mathbf{W}\cdot\mathbf{V}\cos\phi$ is evaluated as an integral over arc length as

$$\int_{\epsilon_{I,J-1}}^{\epsilon_{I,J}} R\mathbf{W}\cdot\mathbf{V}\cos\phi d\epsilon = -R\cos\nu\int_{\epsilon_{I,J-1}}^{\epsilon_{I,J}}\sin\epsilon d\epsilon = R(r_{I,J} - r_{I,J-1}). \tag{3.10}$$

Similarly, the top edge [I:I-1, J] is integrated over arc length as

$$\int_{\delta_{I,J}}^{\delta_{I-1,J}} R\mathbf{W}\cdot\mathbf{U}\cos\phi d\delta = -R\sin\mu\int_{\delta_{I,J}}^{\delta_{I-1,J}}\sin\delta d\delta = R(r_{I-1,J} - r_{I,J}). \tag{3.11}$$




Integrating counter-clockwise around the grid cell, similar formulas for different components yield

$$\Delta \mathbf{A} = -R\Delta t[(\mathbf{P}_{I,J-1} - \mathbf{P}_{I-1,J-1})\Phi_{BOTTOM} + (\mathbf{P}_{I,J} - \mathbf{P}_{I,J-1})\Phi_{RIGHT} +$$
$$+ (\mathbf{P}_{I-1,J} - \mathbf{P}_{I,J})\Phi_{TOP} + (\mathbf{P}_{I-1,J-1} - \mathbf{P}_{I-1,J})\Phi_{LEFT}]/K. \tag{3.12}$$

If all four edge values of $\Phi$ are the same, then cancellation of the $\mathbf{P}$'s causes cell mean $\Delta \mathbf{A}$ to be $\mathbf{0}$.

Relationships like Eq. 3.10 or Eq. 3.11 apply to any great circle arc, not just cubed-sphere edges. To check this: any arc
from $\mathbf{P}_1$ to $\mathbf{P}_2$ is a subset of a great circle that intersects the Equator at two points $\mathbf{I} = (i, j, 0)$ and $-\mathbf{I}$; the longitudinal angular rotation coordinate around this axis is $\xi$ and the latitudinal coordinate from $-\mathbf{I}$ to $\mathbf{I}$ is $\eta$. $\mathbf{P}$ is computed as $(i \sin \eta - j \cos \xi \cos \eta, i \cos \xi \cos \eta + j \sin \eta, \sin \xi \cos \eta)$, and the horizontal unit vector perpendicular to the arc at point $\mathbf{P}$ is $\mathbf{F} = \mathbf{I} \times \mathbf{P}/|\mathbf{I} \times \mathbf{P}|$. $\mathbf{W} \cdot \mathbf{F} = -\sin \xi \sin \eta / \cos \phi$.

$$\int_{\eta_1}^{\eta_2} R\mathbf{W} \cdot \mathbf{F} \cos \phi d\eta = -R \sin \xi \int_{\eta_1}^{\eta_2} \sin \eta d\eta = R(r_2 - r_1). \tag{3.13}$$

Renaming the vertices of a spherical polygon with $N$ arcs $\mathbf{P}_0, \mathbf{P}_1, ... \mathbf{P}_N = \mathbf{P}_0$ counted counter-clockwise around the cell, and generalizing Eq. 3.12:

$$\Delta \mathbf{A} = R\Delta t \sum \Phi_{n-1/2}(\mathbf{P}_{n-1} - \mathbf{P}_n)/K. \tag{3.14}$$

Appendix B shows that if $\Phi$ is a linear function of $\eta$ in the arc, then integrals like Eq. 3.13 can be computed in closed form knowing $\Phi$ at each end of the arc, and Eq. 3.14 can be modified to use $\Phi_n$ and $\Phi_{n-1}$ instead of $\Phi_{n-1/2}$ as shown in Eq.
B4. This is applicable to model IB where primary mass and $\Phi$ are centered at the momentum cell's vertices and $\Phi$ may be interpolated along the edges. For the Section 4 tests performed on model IB, Eq. 3.14 and Eq. B4 produce very similar results. Eq. 3.14 is used because it is simpler.

### 3.3 Advection for model IB

Change in primary cell mass by advection, in flux form, is derived from Green's Theorem and Eq. 2.27.

$$K\Delta h = -\Delta t \sum M_P \tag{3.15}$$

where $M_P$ (kg/s) is the outwardly transported mass at primary cell edges. $M_P$ at each edge is the product of the mass per unit area at the edge, the perpendicular velocity component, and the length of each edge. Using the same notation of variables $\mathbf{I}$, $\eta$, and $\mathbf{F}$ as in the prior subsection, $M_P$ along an edge from $\eta_1$ to $\eta_2$ is computed as

$$M_P = \int_{\eta_1}^{\eta_2} Rh\mathbf{S} \cdot \mathbf{F} d\eta = R \int_{\eta_1}^{\eta_2} h(\mathbf{P} \times \mathbf{S}) \cdot (\mathbf{P} \times \mathbf{F}) d\eta =$$
$$= R \int_{\eta_1}^{\eta_2} h\mathbf{A} \cdot \mathbf{E} d\eta = R \int_{\eta_1}^{\eta_2} h\mathbf{A} \cdot d\mathbf{P} \tag{3.16}$$





where $\mathbf{E}$ is the unit vector parallel to the edge. The final form of Eq. 3.16 shows the elegance of using specific angular momentum as the transport variable; integration around a cell's boundary follows the position vector without a perpendicular velocity computation. As with Eq. 3.14 for the pressure gradient force, $h_{n-1/2}\mathbf{A}_{n-1/2}$ can be the average value over the counter-clockwise arcs, $\mathbf{P}_{n-1}$ to $\mathbf{P}_n$, so that

$$\Delta h = R\Delta t \sum h_{n-1/2}\mathbf{A}_{n-1/2} \cdot (\mathbf{P}_{n-1} - \mathbf{P}_n)/K. \tag{3.17}$$

The inherent symmetry between variables of the shallow water equations is evident by Eq. 3.14 and Eq. 3.17. As shown in Appendix B, variables $\mathbf{A}$ or $h\mathbf{A}$ may also be linearly interpolated with respect to $\eta$ so that the formula for $\Delta h$ is closer to Eq. B4. This is applicable to model IB because $\mathbf{A}$ is located at the primary cell corners. For Section 4 tests performed on IB, Eq. 3.17 is used; the more complex version with linearly interpolated $\mathbf{A}$ produces similar results.

The change in grid cell mean relative angular momentum by advection, in flux form, is equal to the summation of the angular momentum fluxes $\mathbf{V}_T$ (kg m/s$^2$) at the edges multiplied by $\Delta t$. $\mathbf{V}_T$ is the product of $\mathbf{A}$ times $M_T$, the mass flux of momentum cell edges designated by subscript $T$.

$$\mathbf{V}_T = \int Rh\mathbf{A}(\mathbf{S}\cdot\mathbf{F})d\eta = R\int h\mathbf{A}[(\mathbf{P}\times\mathbf{S})\cdot(\mathbf{P}\times\mathbf{F})]d\eta =$$
$$= R\int h\mathbf{A}(\mathbf{A}\cdot\mathbf{E})d\eta = R\int h\mathbf{A}(\mathbf{A}\cdot d\mathbf{P}) = \mathbf{A}M_T. \tag{3.18}$$

If $h_{n-.5}$ and $\mathbf{A}_{n-.5}$ are average values over the arcs surrounding the momentum cells, then the change in angular momentum averaged over a cell is

$$\Delta(h\mathbf{A}) = R\Delta t \sum h_{n-.5}\mathbf{A}_{n-.5}[\mathbf{A}_{n-.5} \cdot (\mathbf{P}_{n-1} - \mathbf{P}_n)]/K. \tag{3.19}$$

For stability purposes, if $\mathbf{A}$ is constant throughout several cells in a neighborhood even though $M_T$ may be irregular, $\Delta\mathbf{A}$ should be $\mathbf{0}$ throughout the neighborhood. If principal mass cells and momentum cells are identical, $M_P = M_T$, this principle is obeyed automatically, assuming that $\mathbf{V}_T = \mathbf{A}M_T$ and $\mathbf{A}$ is the constant.

Primary and momentum cells are not identical for model IB; triangular momentum cells are centered at the corners of primary cells. The solution for momentum cell mass presented here is different from that of Ringler et al. (2010) where $M_P$ and $M_T$ are labeled $F_e$ and $F_e^\perp$ respectively. The area of a momentum cell is the summation of quadrilateral areas inside three touching primary cells, and the momentum cell's mass is the summation of the three quadrilaterals' masses (Figs. 2 or 3). Mass is always uniformly distributed with respect to area in primary cells, and the change in mass in a momentum cell during an advective time step is determined by the change in mass of primary cells. Mass fluxes $M_T$ must be chosen so that their mass changes into a momentum cell match that caused by primary mass changes. $M_T$ is initially computed along an arc between two primary cell centers, but $M_T$ is then modified as explained in the caption to Fig. 3. $M_T$ halves are minimally adjusted so that the change in mass per unit area of each quadrilateral area of a primary cell is identical during the time step. Adjusted $M_T$ halves adjust full $M_T$. This technique satisfies the stability principle of the prior paragraph.





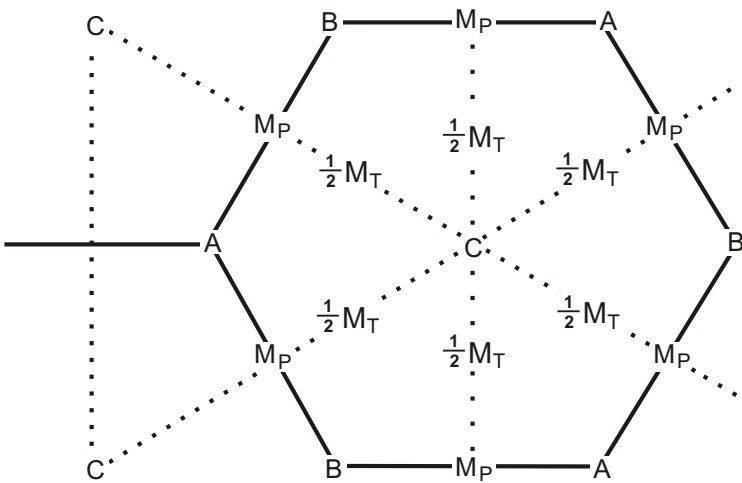

**Figure 3.** Points C are primary cell centers or triangular momentum cell corners. Points A and B are primary cell corners or momentum cell centers. Quadrilaterals are the intersection area between primary and momentum cells. Arc intersection points D are overwritten by $M_P$ in the diagram. D is halfway between C points but not halfway between A and B. Mass flux $M_P$ along a primary cell edge is computed and not changed; it is partitioned to each quadrilateral edge proportional to arc length. Change in mass of a primary cell by advection is computed by summing $M_P$ along the cell's edges. This total change is partitioned into each quadrilateral of the cell proportional to area. The initial value of mass flux $M_T$ is computed along each momentum cell edge; half of $M_T$ is attributed to each primary cell through which it passes. In each primary cell, the half $M_T$'s are adjusted minimally so that the sum of the mass fluxes into each quadrilateral matches the quadrilateral's expected change. Each final $M_T$ is equal to the sum of the two adjusted "half" $M_T$'s.





### 3.4 Coriolis force

For computer implementations of symmetric equations created so far, momentum components have not been defined on staggered locations; all three components reside at the same locations. For each velocity component, the other two components determine the velocity that is perpendicular to the first component. Thus $(pc - ra)/\cos\epsilon$ is the velocity component that

aligns with $\mathbf{P} \times \mathbf{V}$. The Coriolis acceleration (m/s$^2$) acting on $v$ is $2\Omega\sin\phi(pc - ra)/\cos\epsilon$ and the acceleration acting on $b$ is $2\Omega\sin\phi(pc - ra)$. Each specific angular momentum component is accelerated by its local components:

$$\Delta\mathbf{A} = 2\Omega\Delta t\mathbf{A} \times \mathbf{P}\sin\phi. \tag{3.20}$$

### 3.5 Icosahedral models

Starting with 12 vertices and 20 triangles of the icosahedron, new vertices for the raw grid are formed from the center points

of triangular edges. After $m$ iterations of this process the triangular lattice contains $2 + 10 \cdot 4^m$ vertices as described by Stuhne and Peltier (1999). $m$ is called the grid level. These vertices are the centers of primary cells where scalar quantities such as $h$ are defined. Primary cells are mainly irregular hexagons although there are 12 regular pentagons centered on the vertices of the original icosahedron. For model IB, each three nearest primary cell centers form the vertices of a spherical triangle where momentum is defined and which contains the common corner of the three primary cells; these corners designate momentum

cell centers. There are $20 \cdot 4^m$ triangular momentum cells that cover the whole sphere; it is called the "dual" grid by Ringler et al. (2010) and others. See Fig. 2.

Starting from the raw grid, Heikes et al. (2013) reposition and optimize the location of the cell centers producing a tweaked grid. For both the raw and tweaked grids, arc edges of primary cells are the perpendicular bisectors of nearby primary cell centers. The centroid grid's primary cell centers and corners are also repositioned from raw grid locations, but primary cell

edges are neither perpendicular nor bisectors of primary center arcs. For the raw and tweaked grids, cell centers do not coincide with the cell centroid for neither primary nor momentum cells. For the centroid grid, centers and centroids coincide for both primary and momentum cells (see Table 1). The centroid grid here is different from the CVT grid of Du et al. (2003) which had only primary cell coincidence and different from SCB method of Miura and Kimoto (2005) which had only momentum cell coincidence. Each icosahedral model, IB or IK, may use any grid; a capital letter: R = raw, T = tweaked, C = centroid, is

affixed to a model's label that show the grid. IKC works poorly.

Table 1 shows various properties of the three grids for grid levels 4 through 8. A significant property is "Smallest arc length from primary cell corner to D divided by half of ArcA length"; ArcA is between two primary cell corners and D is the intersection of ArcA and the arc between the primary centers. The raw grid value in Table 1 stays at 81% while the tweaked grid converges to unity with increasing resolution. For this reason, IKT is superior to IKR. IB performs edge computations

using only values of the two primary cells, ignoring the fact that D is not the center of ArcA. IK, CSK, and the icosahedral model of Lee and MacDonald (2009) use additional primary cell data to compute cell edge values, but this makes things more difficult when applying such models to an ocean domain.




Considering the most extreme momentum cells in Table 1, the raw grid triangular momentum cells of model IB are more equilateral than are those of the tweaked grid. Momentum cells are more important for model IB, and consequently IBR is as good as IBT. An expanded version of Table 1 with additional parameters is available at https://aom.giss.nasa.gov .

The time scheme is leap-frog initialized every 8 to 10 time steps by forward-backward steps. Alignment of $\mathbf{A}$ (Section 3a)

is performed after every step. The geometry subroutine is complex, but the computational subroutines are simple because angular momentum components are treated identically and use the same lines of code. $M_P$ is computed using 2-point second order differencing. Some momentum computations use upstream differencing. This causes the flow to be stable, but slightly diffusive. However, no other smoothing or filtering subroutines are needed. Computations are performed as locally as possible and consequently the icosahedral models are suitable for an ocean domain or step-mountain atmosphere.

For model IB, the unadjusted $M_T$ is computed using second order differencing, but implementing the stability adjustment (Section 3c) adds complexity to the code. When computing $\mathbf{V}_T = \mathbf{A}M_T$, $\mathbf{A}$ is a mixture of half second-order and half linear upstream advection. Unlike IK and the unfiltered lat-lon B-grid scheme of Arakawa and University of California (1972), model IB seldom generates alternating patterns in the height field. The Fortran source code for IB is available at https://aom.giss.nasa.gov or on Zenodo at http://doi.org/10.5281/zenodo.1313736 .

Model IK uses $h$ at primary grid cell corners interpolated from its value of the three primary cells touching the corner. Different interpolation formulas were tested, but Eq. 3.5a of Heikes et al. (2013) was deemed best.

## 4 Test case results

The following two lat-lon climate models, reduced to one layer for the shallow water equations, were applied to the test cases below. Arakawa's second order B-grid (velocity components defined at primary cell corners) lat-lon model with GISS

ModelE's pole modifications and filters on mass and velocity (Schmidt et al., 2006) is labeled LLB. Arakawa's second order C-grid (velocity components perpendicular to primary cell edges) lat-lon model as modified by Russell (2007) is labeled LLC which also applies a filter to velocity components. The symmetric equations models CSK, IB, and IK do not apply any external filters to mass or momentum other than alignment. CSK performs more 2-point interpolations of variables than do the other models because of its parallelogram shaped cells; this probably explains CSK's poor performance for some of the tests. IB, the

simplest model, has few choices to tune, the main ones being the mixture of second-order versus linear upstream for momentum advection (.5 and .5) and the choice of grid (Raw, Tweaked, or Centroid). The proprietary IK model is more complicated, and the Tweaked grid is superior for this model. Table 2 shows the acronyms and grids of the models used below.

Symmetric equations models IBR, IBT, IBC, IKT, and CSK are represented; IKR and IKC are not, being worse than IKT. Each model was tested with different initial conditions and different horizontal resolutions that approximate $4°$, $2°$, $1°$ and $.5°$;

the symmetric models were tested at $.25°$ in addition. Approximate $1°$ uses 64 primary cells along the triangular edge of an icosahedron, 88 cells along the edge of a cube face, and 180 latitude bands for LL models; for other resolutions these numbers are doubled or halved.



**Table 1.** Properties of the raw, tweaked, and centroid grids; numerical columns are for grid levels 4 through 8. ArcA is arc between adjacent primary cell corners or momentum cell centers. ArcC is arc between nearby primary cell centers or momentum cell corners. Point D is intersection of ArcA and ArcC. Ideal number is 1 for first three properties and 0 for last two properties. Radius is square root of cell area divided by $\pi$.

| Grid | 4° | 2° | 1° | .5° | .25° |
|---|---|---|---|---|---|
| \multicolumn Smallest arc length from primary cell corner to D divided by half of ArcA length | | | | | |
| Raw | .80648 | .80655 | .80656 | .80657 | .80657 |
| Tweaked | .96796 | .98366 | .99174 | .99585 | .99792 |
| Centroid | .87402 | .87354 | .87344 | .87342 | .87341 |
| For most extreme momentum cell: smallest ArcC length divided by largest | | | | | |
| Raw | .85110 | .85076 | .85068 | .85066 | .85065 |
| Tweaked | 83810 | .83617 | .83543 | .83515 | .83504 |
| Centroid | .85091 | .85070 | .85066 | .85065 | .85065 |
| Smallest primary cell area divided by largest | | | | | |
| Raw | .74170 | .73610 | .73468 | .73433 | .73424 |
| Tweaked | .94809 | .95050 | .95206 | .95249 | .95270 |
| Centroid | .36786 | .30111 | .24662 | .20203 | .16550 |
| Largest distance from centroid to center for primary cells divided by cell radius | | | | | |
| Raw | .09710 | .09710 | .09710 | .09710 | .09710 |
| Tweaked | .04250 | .04000 | .03899 | .03865 | .03857 |
| Centroid | .00000 | .00000 | .00000 | .00000 | .00000 |
| Largest distance from centroid to center for momentum cells divided by cell radius | | | | | |
| Raw | .28473 | .28571 | .28595 | .28601 | .28603 |
| Tweaked | .28461 | .28568 | .28594 | .28601 | .28603 |
| Centroid | .00000 | .00000 | .00000 | .00000 | .00000 |

**Table 2.** Acronyms, number of primary cells at 1° resolution, and model description.

| Acronym | # cells | Model description |
|---|---|---|
| IBR | 64*64*10+2 | icosahedral raw grid, momentum cells centered at primary corners |
| IBT | 64*64*10+2 | icosahedral tweaked grid, momentum cells centered at primary corners |
| IBC | 64*64*10+2 | icosahedral centroid grid, momentum cells centered at primary corners |
| IKR | 64*64*10+2 | proprietary icosahedral raw grid |
| IKT | 64*64*10+2 | proprietary icosahedral tweaked grid |
| CSK | 88*88*6 | proprietary gnomonic cubed-sphere grid |
| LLB | 288*178+2 | lat-lon Arakawa B-grid (Schmidt et al., 2006) |
| LLC | 288*178+2 | lat-lon Arakawa C-grid (Russell, 2007) |





### 4.1 Solid body rotation without bottom topography (SBR)

Using the parameters of Test Case 2 of Williamson et al. (1992), a perfect model would maintain the initial mass and velocity fields indefinitely. For a second-order model, the mass and velocity errors should decrease by a factor of 4 when doubling the horizontal resolution. Figure 4 shows the area-weighted root mean square $l_2$ norm of the mass field after 5 days of integration. This Figure may be compared to Fig. 2 of Stuhne and Peltier (1999) and to Fig. 3 of Lee and MacDonald (2009) whose data is averaged over the first 5 days. Both older icosahedral models, LLB, and LLC show error reduction factors of about 4 for doubling resolution. Models IBR and IBC show factors close to 3 that decrease with finer resolution, but increase with less upstream advection of momentum. IKT and CSK show impressive reduction factors of 6 and 5 respectively. As mentioned in the third paragraph of Section 3e, with finer resolution the intersection point D gets closer to the center of primary cell edge arc for model IKT; this increases the precision of this model.

Figure 5 shows the $l_1$ and $l_\infty$ norms as a function of time for the first five days of the $2°$ resolution models IBR, IKT, CSK, and LLC. For all resolutions of the symmetric equations models, the mass field error increases linearly with time; the error after 50 days is nearly ten times that after 5 days. For most resolutions of LLC, the mass field error after 80 days is the same as the error after 5 days. This is also the case for coarsest resolutions, $4°$ and $2°$, of LLB, but $1°$ resolution LLB diverges after 78 days for any reasonable time step and $.5°$ resolution produces polar instabilities although it survives for 100 days and beyond. The present symmetric models of Fig. 5 can be compared to Fig. 8 of Heikes and Randall (1995) that show smaller errors, Fig. 2 of Stuhne and Peltier (1999) that show larger errors, and Fig. 2 of Lee and MacDonald (2009) that is comparable to IBR. The older models show less error growth with increasing time. Both the present $2°$ LLC and the C-grid model of Heikes and Randall at $4°$ resolution show occasional instabilities that do not affect the long term error growth.

In general, the mass field error of IBR is less than that of IBT, often exceeding more than 10% for the coarser resolutions, but closer to 1% for $.25°$ resolution.

### 4.2 Rossby-Haurwitz wave 3 (RH3)

For cubed-sphere models that initialize from repetitive wave 4 initial conditions, each of the 4 wave-lengths lies above the same grid arrangement and copies remain identical when the wave's axis passes through the center of a cube face as is the case for CSK. The same statement applies to icosahedral models with wave 5 initial conditions when the axis passes through a vertex of the icosahedron as is the case for IB and IK. The initial conditions used here are Rossby-Haurwitz wave 3, 3 being relatively prime to both 4 and 5. With wave 3, separate wave-lengths diverge among themselves for models CSK, IB and IK, and there is more variety in the errors that occur. Plots (not displayed) show that CSK for Rossby-Haurwitz wave 4 and IB or IK for Rossby-Haurwitz wave 5 maintain their wave-length shape, but other situations do not. Some results of CSK and IBR for Rossby-Haurwitz waves 3, 4 and 5 are shown at https://aom.giss.nasa.gov under "RHn".

Figure 6 shows horizontal plots of $h + h_S$ after 45 days of integration at $2°$ resolution for six models: IBR, IBC, IKT, CSK, LLB, and LLC. Not displayed is model IBT, which no longer has 3 peaks at $2°$ resolution, but is comparable to IBR for finer resolutions. Although IBC has destroyed the pattern for $2°$ resolution, it actually looks better than IBR for $.5°$ after 45 days.





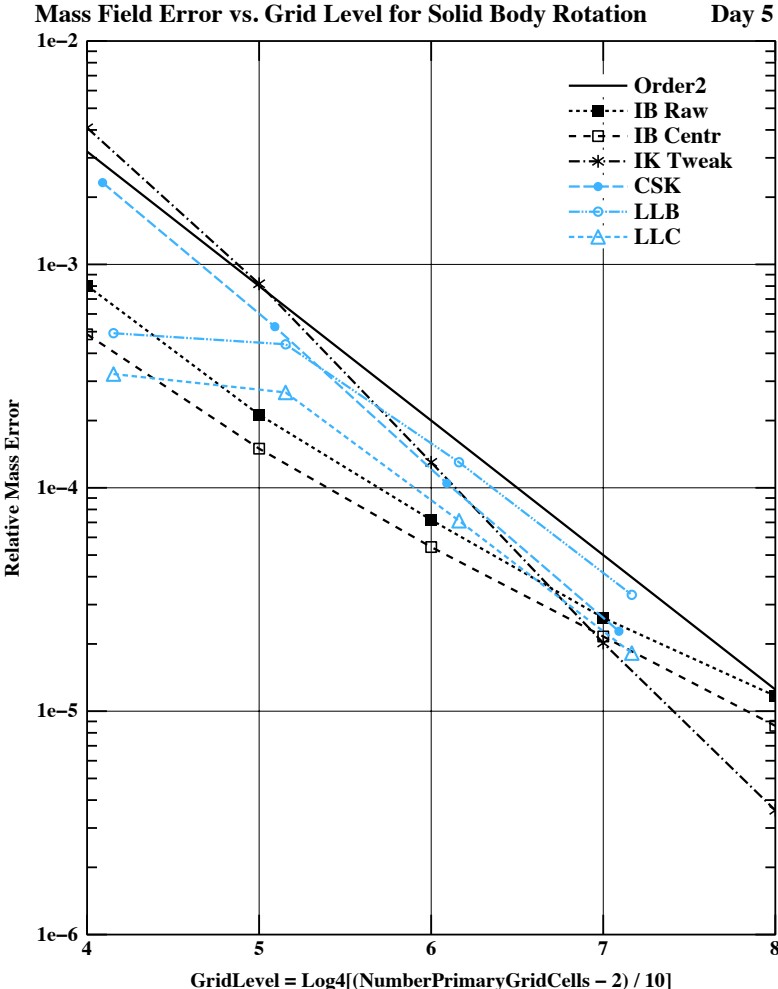

**Figure 4.** Mass field error $l_2$ as function of grid level for solid body rotation initial conditions without bottom topography on day 5. Black line shows ideal mass error: factor of 4 error reduction for doubling horizontal resolution. Models represented are IBR, IBC, IKT, CSK, LLB, and LLC. For last 3 models, grid level is computed as $\log_4[(N-2)/10]$ where $N$ is the number of primary grid cells.

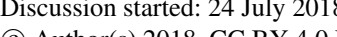





**Figure 5.** Mass field errors $l_1$ and $l_\infty$ as function of time for solid body rotation initial conditions without bottom topography. Models represented are IBR, IKT, CSK, and LLC at $2°$ resolution.





**Figure 6.** Horizontal plots of $h + h_S$ for models IBR, IBC, IKT, CSK, LLB, and LLC at $2°$ resolution after 45 days of integration starting from Rossby-Haurwitz wave 3 conditions.



**Table 3.** Relative change (%) of specific kinetic energy for wave numbers 0 and 3 and total energy for various resolutions and models IBR, IBT, IBC, IKT, CSK, LLB, LLC, and National Center for Atmospheric Research spectral transform model at T42 resolution [Hack and Jakob, 1992] after 40 days of integration. Initial conditions are Rossby-Haurwitz wave 3. "diverge" means simulation diverged.

| Specific kinetic energy change (%) for wave number 0 | | | | | |
|---|---|---|---|---|---|
| Model | 4° | 2° | 1° | .5° | .25° |
| IBR | -19.239 | -5.269 | -1.747 | -.968 | -.428 |
| IBT | -21.529 | -2.017 | -2.042 | -.935 | -.488 |
| IBC | -16.416 | -3.246 | -1.119 | -.587 | -.255 |
| IKT | -12.403 | +7.200 | +1.320 | +.264 | +.063 |
| CSK | -6.002 | +9.298 | +7.031 | +8.202 | -.256 |
| LLB | diverge | -.193 | +.357 | +.019 | |
| LLC | diverge | -.076 | -.009 | +.020 | |
| T42 | +28.448 | | | | |
| Specific kinetic energy change (%) for wave number 3 | | | | | |
| IBR | -74.077 | -44.401 | -22.915 | -10.812 | -5.264 |
| IBT | -74.564 | -76.745 | -29.125 | -10.338 | -5.280 |
| IBC | -69.419 | -69.115 | -49.637 | -6.864 | -3.108 |
| IKT | -98.647 | -50.649 | -10.147 | -1.455 | -.233 |
| CSK | -89.837 | -93.712 | -75.181 | -74.604 | -.825 |
| LLB | diverge | -.061 | +.006 | -.040 | |
| LLC | diverge | +.038 | -.011 | -.049 | |
| T42 | -5.552 | | | | |
| Total energy change (%) | | | | | |
| IBR | -1.3673 | -.6534 | -.3122 | -.1517 | -.0753 |
| IBT | -1.4027 | -.6614 | -.3134 | -.1520 | -.0754 |
| IBC | -1.2850 | -.5972 | -.2467 | -.1055 | -.0476 |
| IKT | -1.9683 | -.6838 | -.1245 | -.0173 | -.0023 |
| CSK | -1.6541 | -.3508 | -.0523 | -.0073 | -.0011 |
| LLB | diverge | -.0001 | -.0055 | -.0002 | |
| LLC | diverge | -.0000 | -.0000 | -.0000 | |
| T42 | +1.0181 | | | | |

CSK performed well for SBR initial conditions, but it is the worst model for RH3, being unable to maintain the RH3 pattern for 45 days except for .25° resolution. Its reduced quality is partially caused by the grid edge lines not being perpendicular. LLB at 2° resolution is beginning to develop alternating patterns in the orange color of Fig. 6, but LLB is excellent for finer resolutions.





Table 3 shows the change in spectral specific kinetic energy for wave numbers 0 and 3 and the change in total energy for models IBR, IBT, IBC, IKT, CSK, LLB, LLC, and National Center for Atmospheric Research spectral transform model (STSWM) at T42 resolution (Hack and Jakob, 1992) after 40 days of integration. Icosahedral and cubed-sphere output was interpolated to a high resolution lat-lon grid before performing spectral decomposition. For each model's finest resolution, the

reduction of wave 3 energy is between 0 and 5.6%. Courser resolutions show greater reductions in wave 3 energy and CSK is less similar to the high resolution results than are the icosahedral models. All of the symmetric models have some amount of upstream advection of momentum which reduces (kinetic and) total energy as shown in Table 3. Model IB has more total energy loss than other models, but it is smoother and more stable.

Except for LLC, which maintains its pattern well for 100 days, the other $1°$ resolution models deviate from the wave 3

pattern after different numbers of days: IBR – 45, IBT – 45, IBC – 37, IKT – 50, CSK – 26, LLB – 63. Both LLB and LLC $4°$ models diverge in the first day of integration, but all other models and resolutions survive for at least 100 days; the final pattern is smooth, but may be unrecognizable.

### 4.3   Initial solid body rotation but with Earth's bottom topography (SBRZ)

The initial velocity is eastward, 50 (m/s) at the equator multiplied by cosine of latitude. The initial global mean height field

top is 10,000 (m) with the latitudinal distribution to maintain itself were there no bottom topography. But $h$ is reduced by the Earth's bottom topography which has peaks as high as 5600 (m). SBRZ is a more realistic and severe than Test Case 5 of Williamson et al. (1992) because it has faster initial velocity, higher mountains, and much larger topography gradients that increase with finer resolution, particularly at the Andes. In Williamson, topography gradients are independent of resolution. Greater accelerations by the pressure gradient force partially explains why the time step is not inversely proportional to the

linear horizontal resolution.

Although the lat-lon models use Fourier polar filters on east-west mass flux and pressure gradient force and other filters on prognostic variables, the symmetric equations models do not. The stability of IB, IK and CSK is maintained by using the proper amount of linear upstream advection of momentum.

For the videos discussed later, the leap-frog time step of each model is interrupted every 8 time steps, and $2\Delta t$ is an integral

division of 900 (s), the video time step. Table 4 shows, for each model and resolution, the largest time step for which the model survives for 50 days without major instabilities, or after what day the model diverged. Time steps were limited to be greater or equal to one third that of IBR for the same approximate resolution. All IB models are stable. IBC requires smaller time steps than IBR or IBT because IBC has some smaller grid cells; this discrepancy increases with resolution as shown in Table 4. IK models are not as stable as IB and require time steps that are half that of IB for finer resolutions. Except for $4°$, LLC

diverges for any time step during the 50 days. LLB diverges for resolutions $4°$ and $.5°$, and requires very short time steps for $1°$ resolution.

Table 5 shows the change in kinetic energy after 50 years of integration for different models and resolutions. For resolutions $1°$ or finer, the kinetic energy for all models that survive is within 5% of the original value. Kinetic energy is not a conserved



**Table 4.** For each model and approximate horizontal resolution, the largest dynamical time step, half of leap-frog time step, in seconds for which the model survives for 50 days without major instabilities, or after what day the model diverged (in parentheses), for solid body rotation initial conditions with Earth's bottom topography.

| Model | 4° | 2° | 1° | .5° | .25° |
|---|---|---|---|---|---|
| IBR | 450 | 225 | 112.5 | 56.25 | 28.125 |
| IBT | 450 | 225 | 112.5 | 56.25 | 28.125 |
| IBC | 225 | 150 | 75 | 37.5 | 15 |
| IKT | 450 | 150 | 56.25 | 30 | 12.5 |
| CSK | 225 | 150 | 90 | 45 | 22.5 |
| LLB | (8) | 150 | 37.5 | (27) | |
| LLC | 225 | (26) | (9) | (8) | |

**Table 5.** Percentage change in kinetic energy for different models and resolutions after 50 days of integration from solid body rotation initial conditions with Earth's bottom topography. "diverge" means simulation diverged.

| Model | 4° | 2° | 1° | .5° | .25° |
|---|---|---|---|---|---|
| IBR | -37.06 | -13.93 | -1.51 | -.97 | -.05 |
| IBT | -41.78 | -15.18 | -3.87 | -.91 | -.77 |
| IBC | -38.42 | -9.34 | -1.43 | -1.22 | +3.83 |
| IKT | -59.35 | -25.17 | -4.72 | +1.12 | +4.32 |
| CSK | -57.12 | -14.00 | -1.75 | +2.25 | +1.73 |
| LLB | diverge | +1.47 | -2.33 | diverge | |
| LLC | -32.86 | diverge | diverge | diverge | |

quantify, but for coarse resolution models, its numerical reduction coincides with the washing out of highs and lows of the height field as shown in the videos discussed subsequently.

Figure 7 shows horizontal plots of $h + h_S$ for resolutions 2° and .5°, and models IBR, IKT, and CSK after 34 days of integration. IBT and IBC are not displayed, but their plots are similar to IBR. Models IKT and CSK display alternating patterns aligned with grid lines in South America. This occurs for both 2° and .5° resolutions, but is more difficult to see for .5° because of the smallness of the printed page. For 2° models shown in Fig. 7, IBR is most similar to the .5° models. There is discrepancy among the non-displayed .25° symmetric models after 50 days; IBR and IKT are more similar to each other than to CSK. Non-displayed LLB diverges at .5° resolution, but at 1°, it is similar to IBR.

Videos of 50 simulated days for models IBR, IKT, CSK, LLB, and LLC are displayable of limited quality on YouTube for all resolutions (https://www.youtube.com/user/NASAGISStv/videos). Label for each video contains model acronym, grid edge





**Figure 7.** Horizontal plots of $h + h_S$ for models IBR, IKT, and CSK and resolutions $2°$ and $.5°$ after 34 days of integration starting from 50 (m/s) eastward velocity multiplied by cosine of latitude, global mean height field top of 10,000 (m) but latitudinal distribution, and Earth's fixed bottom topography, test case 3 SBRZ.



resolution, and "Z", e.g. IBR64Z, CSK88Z, or LLB180Z. IKT models show frequent multi-cell long linear alternating data that may last for several hours. IBR models also show the alternating patterns, but they are much less extensive in occurrence, duration, and spatial distance. CSK is somewhere between IBR and IKT in terms of these alternating patterns. LLB infrequently displays checker-boarding for 3 or 4 adjacent cells. LLC, before running into polar problems, is smooth. At resolutions 1° and

finer and up to 35 days of integration, all of the models that do not diverge are similar to each other.

## 5   Discussion and conclusions

This paper presents symmetric calculus operators on the surface of a sphere that are projected from three-dimensional Cartesian formulas. Symmetric equations are simplified by using specific angular momentum on the unit sphere, **A**, instead of velocity. Components of **A** align with the fixed orthogonal axes whereas the velocity unit vectors **U**, **V** and **W** are not orthogonal and

vary with location. **A** is continuous everywhere, whereas $u$, $v$ and $w$ are discontinuous at their respective poles. **A** simplifies the equation for horizontal velocity **S** (Eq. 2.6) and several formulas in Section 2b. Advection of **A** does not use the metric term, a correction term needed when advection is applied to velocity or linear momentum on the sphere. Applications of Green's Theorem invoke the elegant formulas, Eq. 3.14 for the pressure gradient force and Eqs. 3.17 and 3.19 for advection. All components of relative angular momentum, $h\mathbf{A}$, are conserved without time truncation errors by the flux form advection (ignoring

alignment) used by the symmetric models; LLC conserves the north-south axis component of relative angular momentum by advection, but other components and models lack conservation.

Summarizing the results from the test cases: no one model is clearly superior in all tests for all resolutions and times of integration.

(1) Solid Body Rotation without bottom topography (SBR). For most resolutions, icosahedral models have the lowest relative

mass error at day five of the simulation (Fig. 4). The Arakawa C-grid model, LLC, maintains the same mass field error after 80 days that it had after 5 days, whereas the symmetric equation models all have mass field errors that increase linearly with time. LLC must be considered best for this test.

(2) Rossby-Haurwitz wave 3 (RH3). Large losses of wave number 3 kinetic energy after 40 days (Table 3) show model deficiencies of which CSK is the most egregious. LLC maintains the wave 3 shape longer than the other models (Fig. 6)

and again is the best. For Rossby-Haurwitz wave 4, CSK maintains the wavelength-shape better than does IBR, while for Rossby-Haurwitz wave 5, IBR is superior to CSK for longer integrations (https://aom.giss.nasa.gov under "RHn").

(3) Initial Solid Body Rotation with Earth's topography (SBRZ). Polar problems cause the Lat-Lon models, LLB and LLC, to diverge for different resolutions including .5°. IKT and CSK to a lesser extent produce linear alternating patterns that are much diminished in IBR (Fig. 7 and the videos). IBR and IBT use a longer time step (Table 4) than do other models, and, after

34 days, 2° IBR is more similar to the higher resolution .5° models than are 2° IKT and CSK (Fig. 7). The icosahedral B-grid models, IBR and IBK, are best.

The SBRZ test is the most difficult, but the most realistic test. Given that these one layer models will be the basis for multi-layer climate models with realistic topography, IB models must be considered the best overall. Quadrilateral shaped grid cells





in CSK with non-perpendicular grid line edges lasting over large swaths of the globe causes a systematic error in numerical flow, most noticeably evident in the RH3 test case. Weller et al. (2012) also conclude that "the hexagonal icosahedron gives the most accurate results" for several test cases. Although IKT's reduction factor of 6 for doubling resolution with SBR initial conditions is impressive, it generates frequent alternating linear patterns for SBRZ conditions as shown on the YouTube videos.

The Williamson et al. (1992) test cases are inadequate for one-layer models that will be expanded to climate models simulating Earth; Williamson's Test Case 5 is much less demanding than using SBRZ. Grid imprinting errors in IB and IK are comparable to errors reported by other researchers Stuhne and Peltier (1999); Lee and MacDonald (2009); the errors are not worse than would occur using two horizontal velocity components.

The symmetric equations models presented here use the smallest sensible grid cell stencil needed for a computation. Enlarg-

ing the stencil (as used by Lee and MacDonald [2007]) may improve the results for tests SBR and RH3 by using fourth-order differencing or other methods, but in ocean domains or step-mountain atmospheres, large stencils require many different formulas for flow near ocean coast lines, based on their shape, and each formula is different from that used in the interior. CSK, with its parallelogram shaped cells, has more difficulty in conforming to an ocean domain than does the icosahedral models.

As noted in Section 3e, IB uses only two adjacent primary cell centers when performing computations on their common

edge, even though point D is not the center of the primary cell edge. The computational subroutines of IB, and to a lesser extent IK, are extremely simple, unlike CSK which requires frequent interpolation of variables.

Flux form velocity, represented by two horizontal components, has significant problems where the coordinates become discontinuous. An improvement has been to use forms of the shallow water equations where scalar quantities such as potential vorticity, specific kinetic energy, divergence, stream function, and velocity potential are continuous over the whole sphere and

from which the local horizontal velocity can be resurrected, or integrated using the manipulated scalar quantities. Deficiencies of these methods are complexity of understanding and computer coding. This paper presents another method: vector angular momentum is continuous over the whole sphere and its application via the symmetric equations is simpler than using velocity. Each component of relative angular momentum is conserved by flux form advection without discontinuities. Further work is needed to determine the practical advantages that one scheme may have over others.

*Code availability.*  Fortran source code for model GISS:IB is available at https://aom.giss.nasa.gov or
on Zenodo at http://doi.org/10.5281/zenodo.1313736 .





## Appendix A: Equivalence between new and old forms for $\nabla h$

$$\mathbf{W} = (-q,p,0)/\sqrt{p^2+q^2} = (-\sin\lambda,\cos\lambda,0) \tag{A1}$$

$$\mathbf{N} = (-rp,-qr,p^2+q^2)/\sqrt{p^2+q^2} = (-\cos\lambda\sin\phi,-\sin\lambda\sin\phi,\cos\phi) \tag{A2}$$

$$\partial\lambda/\partial\delta = \mathbf{W}\cdot\mathbf{P}\times\mathbf{U}/\cos\phi = -\sin\lambda/\cos\phi\cos\delta \tag{A3}$$

$$\partial\lambda/\partial\epsilon = \mathbf{W}\cdot\mathbf{P}\times\mathbf{V}/\cos\phi = \cos\lambda/\cos\phi\cos\epsilon \tag{A4}$$

$$\partial\phi/\partial\delta = \mathbf{N}\cdot\mathbf{P}\times\mathbf{U} = \mathbf{W}\cdot\mathbf{U} = -\cos\lambda\sin\phi/\cos\delta \tag{A5}$$

$$\partial\phi/\partial\epsilon = \mathbf{N}\cdot\mathbf{P}\times\mathbf{V} = \mathbf{W}\cdot\mathbf{V} = -\sin\lambda\sin\phi/\cos\epsilon \tag{A6}$$

$$
\begin{aligned}
\nabla h &= \left(\cos\delta\frac{\partial h}{\partial\delta},\cos\epsilon\frac{\partial h}{\partial\epsilon},\cos\phi\frac{\partial h}{\partial\phi}\right)/R = \\
&= \left[\cos\delta\left(\frac{\partial\lambda}{\partial\delta}\frac{\partial h}{\partial\lambda}+\frac{\partial\phi}{\partial\delta}\frac{\partial h}{\partial\phi}\right),\cos\epsilon\left(\frac{\partial\lambda}{\partial\epsilon}\frac{\partial h}{\partial\lambda}+\frac{\partial\phi}{\partial\epsilon}\frac{\partial h}{\partial\phi}\right),\cos\phi\frac{\partial h}{\partial\phi}\right]/R = \\
&= \left(-\sin\lambda\frac{\partial h}{\partial\lambda}/\cos\phi-\cos\lambda\sin\phi\frac{\partial h}{\partial\phi},\cos\lambda\frac{\partial h}{\partial\lambda}/\cos\phi-\sin\lambda\sin\phi\frac{\partial h}{\partial\phi},\cos\phi\frac{\partial h}{\partial\phi}\right)/R = \\
&= \left(\mathbf{W}\frac{\partial h}{\partial\lambda}/\cos\phi+\mathbf{N}\frac{\partial h}{\partial\phi}\right)/R = \left(\mathbf{W}\frac{\partial h}{\partial\lambda}+\mathbf{N}\cos\phi\frac{\partial h}{\partial\phi}\right)/R\cos\phi
\end{aligned}
\tag{A7}
$$

## Appendix B: Closed form integration along an arc

Using the notation for $\mathbf{F}$, $\eta$ and $\xi$ of Section 3.b:

$$\mathbf{W}\cdot\mathbf{F} = -\sin\xi\sin\eta/\cos\phi; \tag{B1}$$

$$r = \sin\xi\cos\eta. \tag{B2}$$

Assume that $\Phi$ were not constant throughout the arc, but instead were a linearly function of $\eta$ from $\Phi_1$ to $\Phi_2$. Then the arc integral of Eq. 3.13 is now

$$
\begin{aligned}
\int R\Phi\mathbf{W}\cdot\mathbf{F}\cos\phi\,d\eta &= \int_{\eta_1}^{\eta_2} R\left(\frac{\Phi_1\eta_2-\Phi_2\eta_1}{\eta_2-\eta_1}+\eta\frac{\Phi_2-\Phi_1}{\eta_2-\eta_1}\right)\mathbf{W}\cdot\mathbf{F}\cos\phi\,d\eta = \\
&= -R\sin\xi\int_{\eta_1}^{\eta_2}\left(\frac{\Phi_1\eta_2-\Phi_2\eta_1}{\eta_2-\eta_1}+\eta\frac{\Phi_2-\Phi_1}{\eta_2-\eta_1}\right)\sin\eta\,d\eta = \\
&= R(\Phi_2 r_2-\Phi_1 r_1)-R\frac{\Phi_2-\Phi_1}{\eta_2-\eta_1}\sin\xi(\sin\eta_2-\sin\eta_1) = \\
&= R(\Phi_2 r_2-\Phi_1 r_1)-R\frac{\Phi_2-\Phi_1}{\eta_2-\eta_1}2\sin\xi\cos\left(\frac{\eta_1+\eta_2}{2}\right)\sin\left(\frac{\eta_2-\eta_1}{2}\right) = \\
&= R(\Phi_2 r_2-\Phi_1 r_1)-R\frac{\Phi_2-\Phi_1}{\eta_2-\eta_1}2 r_C\sin\left(\frac{\eta_2-\eta_1}{2}\right)
\end{aligned}
\tag{B3}
$$





where $r_C$ is the value of $r$ at the center of the arc from $\eta_1$ to $\eta_2$. As the resolution of the grid becomes finer, $r_C$ approaches $(r_1 + r_2)/2$ and $\sin[(\eta_2 - \eta_1)/2]$ approaches $(\eta_2 - \eta_1)/2$, and the integral of Eq. B3 approaches $R(r_2 - r_1)(\Phi_1 + \Phi_2)/2$, which is the result of Eq. 3.13.

Using Eq. B3 instead of Eq. 3.13, the formula of Eq. 3.14 is replaced with

$$5 \quad \Delta \mathbf{A} = R\Delta t \sum \left[ \Phi_{n-1}\mathbf{P}_{n-1} - \Phi_n\mathbf{P}_n + 2\frac{\Phi_n - \Phi_{n-1}}{\eta_n - \eta_{n-1}}\sin\left(\frac{\eta_n - \eta_{n-1}}{2}\right)\mathbf{P}_{n-.5} \right]/K. \tag{B4}$$

*Acknowledgements.* The origins of this paper came from discussions with Maxwell Kelley in 2012 and 2013. He has made contributions to the present manuscript, and models IK and CSK are based on his ideas that may be published in the future. The authors thank Ross Heikes of Colorado State University for the tweaked grid locations. Rainer Bleck was helpful with knowledge of other icosahedral models. Robert Schmunk assisted in creating the YouTube video files. The paper was significantly improved by the editorial process of *Monthly Weather*
10   *Review*.

Climate modeling at GISS is supported by the NASA Modeling, Analysis, and Prediction program, and resources supporting this work were provided by the NASA High-End Computing Program through the NASA Center for Climate Simulation at Goddard Space Flight Center.



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
