# Peer review of "Symmetric Equations on the Surface of a Sphere as Used by Model GISS:IB"

_Geoscientific Model Development, 2018_

## Referee Comment (RC1) · Anonymous Referee #1 · 27 Sep 2018

The idea of presenting symmetrical equations of motion with dependent dynamical variables (more velocity fields than spatial dimensions) is an interesting one insofar as it permits to avoid singularities of the coordinate system.

After reading this manuscript, I was however left with mixed feelings about the approach. It does not seem that the authors found strikingly interesting or advantageous behaviours associated with their approach. In particular, I would have liked to see explained specific features of the numerical solutions that are especially attractive and specific to their approach. Getting rid of singularities may be achieved in different ways. What is particularly attractive in the authors' approach?

I appreciated the thoroughness of the mathematical description that the authors presented (there is one exception, see below). I am confident the mathematical description

presented will allow other scientists to reproduce their approach.

I believe the manuscript would benefit from adding a sub-section on the particularities of the numerical solutions obtained from the symmetrical approach compared to other means of mapping the sphere without singularities (but with discontinuities of the co-ordinates). In other words, the authors should help the reader understand why it may be beneficial to learn the symmetrical approach. Does the increased mathematical complexity worth the effort?

My recommendation is therefore: acceptable with minor revisions.

I also provide this list of minor comments:

p.2 l.1: At this point, please define what is precisely meant by symmetric formulas and isodirectional flow;

p.2, l.5-10: Another possibility, keeping the lat-lon paradigm, is to use the Yin-Yang grid (see Qaddouri et al.). This is operational in Canada and this approach should be mentioned here as well as the other approaches;

p.2, l.9: How are the eight corners of the cubed-sphere singularities? At these points, the determinant of the metric tensors corresponding to each connected domain do not vanish.

p.2, l.25: Isn't the symmetry broken when rotation is introduced? Are the three Cartesian coordinates inertial?

General comments on the introduction: The Introduction should perhaps be shorter and more focused. It is somewhat dry.

p.4, l.26-27: "Three horizontal velocity components ... rotate around each respective axis." Please rephrase. It is unclear how a velocity component can rotate around an axis.

p.5, l.8: Eq. 2.5 should be better justified.

[Figure]

p.25, l.32: Change "years" for "days".
* * *

---

## Referee Comment (RC2) · Anonymous Referee #2 · 28 Sep 2018

**Review of "Symmetric Equations on the Surface of a Sphere as Used by Model GISS:IB"**

**Manuscript ID:**     gmd-2018-126

**Title:**     Symmetric Equations on the Surface of a Sphere as Used by Model GISS:IB

**Authors:**     Gary L. Russell, David H. Rind, and Jeffrey Jonas

**Recommendation:**     accept after minor revisions

**Summary:**

This study propose a new methodology to represent two-dimensional flow on a sphere. Reinterpretting previous studies, the approach in this study use specific angular momentum on the unit sphere and avoid problems associated with singularities in effect. The authors provided applications for some vector calculus and the shallow water equations. They also performed standard test simulations for the shallow water equations and compared with other schemes.

The manuscript is generally well written and organized, though some typos and lacks of descriptions are found. The presented methodology gives the concise and elegant representation of the shallow water equations on a sphere, and it is valuable for the GMD readers. Although the practical advantage of the proposed representation are not evaluated, it seems suitable that this point be addressed in further studies.

Therefore, I recommend the acceptance after minor revisions.

**Comments:**

1. Page 2, Line 11:

   Replace "Putman and Lin" with "Putman and Lin (2007)."

2. Page 3, Line 4–6:

   It is easier to read if what among/between the similarities and differences are discussed is specified.

3. Page 3, Line 17:

   Replace "[]" with "()."

4. Page 5, Line 6–8:

   Can we derive Eq. (2.5) only from the forementioned relationships?

5. Page 6, Eq. (2.11)

   Place a period at the end of the line.

6. Page 6, Line 10:

   Something should be missed in the last sentence.

7. Page 6, Line 13–14:

   The numbering of the faces are not shown in Fig. 1. Additionally, I am not sure that the way of description of order is common, whereas I can guess it.

8. Figure 2:

   Please specify what the grid level 2.

9. Page 20, Line 7–8:

   Please add descriptions about the reasons for the observed facts.

10. Page 20, Line 19:

   Please insert the year number for the reference.

---

## Author Comment (AC1) · 12 Oct 2018

The idea of presenting symmetrical equations of motion with dependent dynamical variables (more velocity fields than spatial dimensions) is an interesting one insofar as it permits to avoid singularities of the coordinate system.

After reading this manuscript, I was however left with mixed feelings about the approach. It does not seem that the authors found strikingly interesting or advantageous behaviours associated with their approach. In particular, I would have liked to see explained specific features of the numerical solutions that are especially attractive and specific to their approach. Getting rid of singularities may be achieved in different ways. What is particularly attractive in the authors' approach?

Relating to non-icosahedral grids Page 2, Lines 12-13: "A problem with the above mentioned grids is that the treatment and behavior at grid edges differs significantly from that away from the edges. Choices that must be made in the pursuit of consistency, have the potential for inducing edge errors. This will not be the case for the approach presented here."

And Lines 16-17: "Although irregularities are distributed all over the sphere, the hope is that icosahedral grid errors are less concentrated than edge errors of other grids and also less severe."

Perhaps what is most attractive is the simplicity of the dynamical subroutines. This point is now made explicitly in the penultimate paragraph of the Conclusions: "Except for the stability requirement of mass fluxes entering momentum cells (Section 3c and Fig. 3), the computational subroutines of IB are extremely simple. There are not separate lines of code for angular momentum; all components use the same lines."

The code is published on zenodo. Manuscript changed.

I appreciated the thoroughness of the mathematical description that the authors presented (there is one exception, see below). I am confident the mathematical description presented will allow other scientists to reproduce their approach.

I believe the manuscript would benefit from adding a sub-section on the particularities of the numerical solutions obtained from the symmetrical approach compared to other means of mapping the sphere without singularities (but with discontinuities of the coordinates). In other words, the authors should help the reader understand why it may be beneficial to learn the symmetrical approach. Does the increased mathematical complexity worth the effort?

We are not in possession of shallow-water models other than those used in the comparisons with other models that have been published for Williamson Test Case 2; and other models can compare with symmetric models that will be published in the present manuscript.

In addition to their reduction of edge errors, a benefit of symmetric equations on an icosahedral grid is the significant reduction of mathematical complexity. In lat-lon codes, eastward and northward velocity components have very different formulas. In symmetric equation codes, a line of code using angular momentum is used for all three components. The geometry subroutine is complicated, but the dynamical subroutines are simple for model IB. Manuscript changed as indicated on the first page of this "Response Letter".

My recommendation is therefore: acceptable with minor revisions.

I also provide this list of minor comments:

p.2 l.1: At this point, please define what is precisely meant by symmetric formulas and isodirectional flow;

If one looks at any component of velocity or angular momentum in most formulas, there are symmetric terms for the other two components as well. [No change to manuscript on this point.]

The phrase "which means the flow is not isodirectional" is removed from manuscript.

p.2, 1.5-10: Another possibility, keeping the lat-lon paradigm, is to use the Yin-Yang grid (see Qaddouri et al.). This is operational in Canada and this approach should be mentioned here as well as the other approaches;

The Yin-Yang grid is now discussed in the second sentence of the third paragraph of the Introduction and is cited in the References. Manuscript modified.

p.2, 1.9: How are the eight corners of the cubed-sphere singularities? At these points, the determinant of the metric tensors corresponding to each connected domain do not vanish.

"eight corner singularities" is replaced with "eight ill-behaved corners" in the manuscript.

p.2, l.25: Isn't the symmetry broken when rotation is introduced? Are the three Cartesian coordinates inertial?

Even the Coriolis subroutine uses the same line of code for all three components. [No change to manuscript.]

The Cartesian coordinates relate to the fixed Earth which is now stated in the last sentence of the Introduction. Manuscript modified.

General comments on the introduction: The Introduction should perhaps be shorter and more focused. It is somewhat dry.

The Introduction is not any shorter, but the main former body of the paragraph starting "The approach here .." on Page 2, Line 26 has been moved to the beginning of Section 2. Manuscript modified.

p.4, l.26-27: "Three horizontal velocity components ... rotate around each respective axis." Please rephrase. It is unclear how a velocity component can rotate around an axis.

As stated at several locations including the Abstract, one of the three components is eastward velocity that rotates around the north-south axis. Understanding this helps to comprehend the other two components that rotate around the equatorial axes. [No change to manuscript.]

p.5, l.8: Eq. 2.5 should be better justified.

The more easily understood formula for A in Eq. 2.5 is now mentioned first. The more complicated formula for S in Eq. 2.6 is simply computed as AxP. Paragraphs 2 and 3 of Section 2.1 are appropriately modified in the manuscript.

p.25, 1.32: Change "years" for "days". Manuscript modified.

---

## Author Comment (AC2)

Response to Reviewer 2

Review of "Symmetric Equations on the Surface of a Sphere as Used by Model GISS:IB"

Manuscript ID: gmd-2018-126
Title: Symmetric Equations on the Surface of a Sphere as Used by Model GISS:IB
Authors: Gary L. Russell, David H. Rind, and Jeffrey Jonas
Recommendation: accept after minor revisions

Summary:

This study propose a new methodology to represent two-dimensional flow on a sphere. Reinterpretting previous studies, the approach in this study use specific angular momentum on the unit sphere and avoid problems associated with singularities in effect. The authors provided applications for some vector calculus and the shallow water equations. They also performed standard test simulations for the shallow water equations and compared with other schemes.

The manuscript is generally well written and organized, though some typos and lacks of descriptions are found. The presented methodology gives the concise and elegant representation of the shallow water equations on a sphere, and it is valuable for the GMD readers. Although the practical advantage of the proposed representation are not evaluated, it seems suitable that this point be addressed in further studies.

Therefore, I recommend the acceptance after minor revisions.

Comments:
1. Page 2, Line 11:
Replace "Putman and Lin" with "Putman and Lin (2007)."
Manuscript modified.

2. Page 3, Line 4–6:
It is easier to read if what among/between the similarities and differences are discussed is specified.
Similarities now state "(he again uses u cos phi)" and differences are stated in the next sentence. Manuscript modified.

3. Page 3, Line 17:

Replace "[]" with "()."
Manuscript modified appropriately.

4. Page 5, Line 6–8:
Can we derive Eq. (2.5) only from the forementioned relationships?
The more easily understood formula for A in Eq. 2.5 is now mentioned first. The more complicated formula for S in Eq. 2.6 is simply computed as AxP. Paragraphs 2 and 3 of Section 2.1 are appropriately modified in the manuscript.

5. Page 6, Eq. (2.11)
Place a period at the end of the line.
Manuscript modified.

6. Page 6, Line 10:
Something should be missed in the last sentence.
"=" is replaced with "is". Manuscript modified.

7. Page 6, Line 13–14:
The numbering of the faces are not shown in Fig. 1. Additionally, I am not sure that the way of description of order is common, whereas I can guess it.
The legend of Fig. 1 states that the upper diagram relates to north polar Face 1; faces are numbered in the lower diagram. [No change to manuscript.]

8. Figure 2:
Please specify what the grid level 2.
"Grid level" is defined in the second sentence of Section 3.5. Although Fig. 2 is first mentioned in the last sentence of Section 2.1, there is no necessity to discuss "grid level" at this point. [No change to manuscript.]

9. Page 20, Line 7–8:
Please add descriptions about the reasons for the observed facts.
The desired explanations are added to the final paragraph of Section 4.1: "All of the symmetric equation models use some amount of upstream advection of momentum. This causes a continual loss of total energy and of structure to the prognostic variables, and is the reason for the continual degradation of the l2 norms in time. The Arakawa B grid and C grid schemes are designed to approximately conserve total energy." Manuscript modified.

10. Page 20, Line 19:
Please insert the year number for the reference.

Manuscript modified.